# CAN TRANSFORMERS CAPTURE SPATIAL RELATIONS BETWEEN OBJECTS?

**Chuan Wen**[1,3,4], **Dinesh Jayaraman**[2], **Yang Gao**[1,3,4,†]
[1]Institute for Interdisciplinary Information Sciences, Tsinghua University
[2]University of Pennsylvania
[3]Shanghai Artificial Intelligence Laboratory
[4]Shanghai Qi Zhi Institute

## ABSTRACT

Spatial relationships between objects represent key scene information for humans to understand and interact with the world. To study the capability of current computer vision systems to recognize physically grounded spatial relations, we start by proposing precise relation definitions that permit consistently annotating a benchmark dataset. Despite the apparent simplicity of this task relative to others in the recognition literature, we observe that existing approaches perform poorly on this benchmark. We propose new approaches exploiting the long-range attention capabilities of transformers for this task, and evaluating key design principles. We identify a simple "RelatiViT" architecture and demonstrate that it outperforms all current approaches. To our knowledge, this is the first method to convincingly outperform naive baselines on spatial relation prediction in in-the-wild settings. The code and datasets are available in https://sites.google.com/view/spatial-relation.

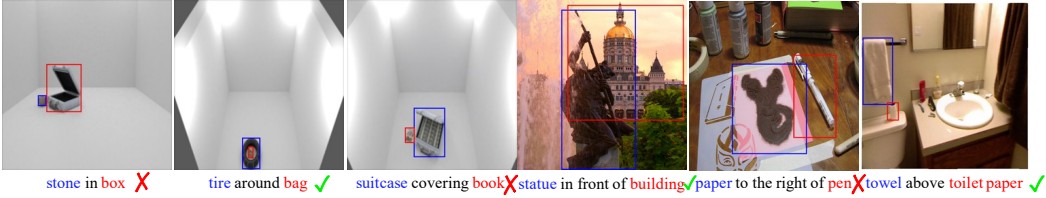

stone in box ✗   tire around bag ✓   suitcase covering book✗ statue in front of building✓ paper to the right of pen✗ towel above toilet paper ✓

Figure 1: Illustrations of spatial relation prediction (SRP) task on Rel3D and SpatialSense+ datasets.

## 1 INTRODUCTION

The spatial relationships between objects are crucial for visual scene understanding. For example, humans understand that "the saucer is **under** the coffee cup", so we can make reasonable manipulation plans: move the cup before picking up the saucer. How well can current computer vision systems complete such a spatial relation recognition task? Given an image and bounding boxes of two queried instances (subject and object), the task is to recognize the spatial relationships. Modern computer vision systems have been widely benchmarked to demonstrate progress along various axes of the visual understanding problem such as scene and object category recognition, object segmentation, etc., but to a lesser extent for understanding object relationships, particularly in the precise physically grounded sense outlined above.

Many prior works have studied *semantic* object relationships (Lu et al., 2016; Zellers et al., 2018; Tang et al., 2019; Li et al., 2021a), including those tied to the spatial orientations of objects. However, such relations are not always physically grounded. For example, in this semantic sense, a picture that is only attached horizontally to a wall is "on a wall", but so is a cat that is walking on top of a wall. These relationships reflect linguistic conventions rather than precise physically grounded spatial relationships between objects such as the cup atop the saucer above. These examples demonstrate that such semantic relationships can be ambiguous and do not necessarily capture the physical

---

† Corresponding author.

properties of the objects involved. Indeed, this shortcoming has been noticed in prior works, where it has been shown that just names of the categories of the objects involved, with no image information at all, already predict such relationships well (Liang et al., 2019; Yang et al., 2019).

Detecting such semantic relationships does not suffice to inform a robot aiming to pick up the saucer mentioned in the opening paragraph. That requires a more precise and physically grounded notion of spatial relationships. To study such relations in-the-wild, we first set up a new benchmark. Two existing datasets Rel3D (Goyal et al., 2020) and SpatialSense (Yang et al., 2019) also target spatial relation prediction. However, as we will show in Section 3.2, Rel3D primarily handles synthetic scenarios with clean backgrounds and textures, and SpatialSense suffers from annotation inconsistencies caused by relation definition ambiguity and language bias, etc. To fix this, we first propose precise, unambiguous, definitions for various commonly used spatial relations in a manner that permits consistent assignments of target labels in complex real-world scenarios. We then relabel the SpatialSense dataset to be consistent with our definitions. Figure 1 shows some samples from this new dataset, SpatialSense+. We use these two datasets, Rel3D (synthetic) and SpatialSense+ (realistic) to comprehensively benchmark spatial relation prediction approaches.

How should we address the **spatial relation prediction (SRP)** task? This task may appear simple at first glance relative to more complex visual recognition tasks (He et al., 2016; 2017; Long et al., 2015) that have already been shown to be achievable with modern computer vision systems. Indeed, what makes this task particularly interesting to study is that despite its apparent simplicity, prior works have consistently reported that no existing computer vision approaches are able to outperform a naive "bbox-only" baseline – making predictions only based on the bounding box coordinates (Goyal et al., 2020; Yang et al., 2019). The existing methods are in very similar paradigms: extracting local features of the two objects from a frozen CNN backbone as input to an MLP relation classifier. Note these architectural choices; while standard for many vision tasks, they make it difficult to explicitly model long-range interactions between objects, as might be necessary for SRP.

If this architecture mismatch is indeed the root cause of the difficulty of SRP with modern computer vision approaches, how might we overcome it? Transformer architectures (Vaswani et al., 2017) have recently achieved great success across various fields, e.g., NLP (Devlin et al., 2019) and computer vision (Dosovitskiy et al., 2021). Unlike CNNs with first-order convolutional operations between images and weights, the Vision Transformer (ViT) (Dosovitskiy et al., 2021) takes advantage of the attention mechanism (Vaswani et al., 2017) to fuse features among image patches. This explicitly introduces the inductive bias about pair-wise relationships and comparison between the image patches, suggesting that transformer-based solutions may be particularly well-suited for SRP.

In this work, we systematically study several carefully designed transformer-based architectures on our precise and physically grounded spatial relation prediction benchmark, comparing them with the best existing CNN-based methods, and identifying a clear winning design, which we call "RelatiViT". Our experimental results demonstrate that RelatiViT significantly outperforms all the existing methods and is the **first** to convincingly use *visual* information to improve performance on this task beyond just relying on the 2D spatial coordinates of the objects. Our experiments further show that even the most advanced large-scale Vision Language Models, such as GPT-4V, Gemini, LLaVA, and MiniGPT-4, fail to achieve satisfactory results on our benchmark, highlighting that the spatial relation prediction is an essential and challenging task for visual reasoning.

## 2 RELATED WORK

**Spatial relationship.** Recognizing the relations between the objects in an image is a challenging problem beyond object recognition and detection in computer vision. Visual relation detection is a well-known task to predict the semantic relations between objects and generate a scene graph from an image. The commonly used benchmarks VRD (Lu et al., 2016), Visual Genome (Krishna et al., 2017) and Open Images (Kuznetsova et al., 2020) are reported to suffer from severe language bias (Tang et al., 2020b; Zellers et al., 2018; Liang et al., 2019) and ill-defined evaluation metrics (Yang et al., 2019). As a result, the existing visual relation detection methods mainly focus on improving the information aggregation on the graph (Xu et al., 2017; Zellers et al., 2018; Tang et al., 2019; Lin et al., 2020; Li et al., 2021a; Lin et al., 2022), data debiasing (Tang et al., 2020b; Chen et al., 2019; Li et al., 2022), and utilizing commonsense (Zareian et al., 2020) or language priors(Lu et al., 2016; Yao et al., 2021). Spatial relationship is also leveraged in some recent robotics works for task planning, but they either only work on limited domains (Bobu et al., 2022; Wang et al., 2022; Ku et al., 2023) or require access to point clouds (Liu et al., 2023b), indicating the

difficulty of recognizing spatial relations from 2D vision representations. Two recent works propose the Rel3D (Goyal et al., 2020) and SpatialSense (Yang et al., 2019) datasets to place attention on the physically grounded spatial relation prediction task from 2D images.

Moving to *methods* for SRP, the above works employ RoI Align to extract the visual features of objects from a CNN encoder, and then feed them into a prediction head (based on message passing (Xu et al., 2017) or LSTM (Zellers et al., 2018; Tang et al., 2019)) to classify the relationship between them. SorNet (Yuan et al., 2022) assumes that object canonical views (pictures taken with the objects facing the camera) are available as queries, relying on them to localize the objects and extract object embeddings before predicting relations. This assumption is impractical in in-the-wild settings with unknown objects. Moreover, their datasets explore only simple cuboid-like geometries of "objects", uncluttered scenes with limited occlusions, fixed tabletop scene layouts, etc. We study a more challenging and realistic setting with real images, complex scene layouts and arbitrary camera pose.

**Relation extraction.**  Relation extraction is a widely studied task in other fields of deep learning. In graph learning, the relation extraction task is to predict the existence of edges between nodes (Zhang & Chen, 2018) or the attributes of the edges (Onuki et al., 2019; Cui et al., 2021). In natural language processing, there are many existing works about sentence-level (Zhang et al., 2017c; Peng et al., 2020) and document-level relation extraction (Tang et al., 2020a; Wang et al., 2019; Zhou et al., 2021; Tan et al., 2022). Here, Wang et al. (2019); Zhou et al. (2021); Tan et al. (2022) show that extracting relation features from a transformer pretrained by BERT (Devlin et al., 2019) is more effective than the graph-based methods (Tang et al., 2020a; Nan et al., 2020). Similarly, we propose to extract the relation features from a pretrained ViT for visual relation prediction tasks.

**Vision Transformer.**  Building on the success of transformers in NLP (Vaswani et al., 2017; Devlin et al., 2019), Vision Transformer (ViT) is proposed to handle image data (Dosovitskiy et al., 2021). Taking advantage of long horizon memory properties of the attention mechanism, ViT has a larger perceptual field and thus excels across various computer vision tasks, e.g., object recognition (Touvron et al., 2021a), detection (Li et al., 2021b) and semantic segmentation (Strudel et al., 2021; Xu et al., 2022), etc. ViT utilizes the attention mechanism to explicitly model pair-wise relationships between image patches, which inspires us to explore *object* spatial relation prediction from pretrained ViTs (Caron et al., 2021; He et al., 2022; Chen et al., 2021; Zhou et al., 2022; Radford et al., 2021).

## 3 PRELIMINARIES

### 3.1 PROBLEM FORMULATION

Consider the example "statue in front of building" in Figure 1, the spatial relation prediction (**SRP**) task expects the model to recognize that the predicate "in front of" is true because the depth of the subject "statue" is smaller than the object "building". Because there might be multiple relations holding simultaneously between the subject and object (multi-label property), we formalize the SRP task as a **K-way binary classification**: given an image $I$, two bounding box coordinates of "subject" and "object" $b_s, b_o \in \mathbb{R}^4$, corresponding object categories $c_s, c_o \in \mathcal{C}$, and a relation predicate $r \in \{0, \cdots, K-1\}$, the $r$-th prediction head $f_\theta^r$ of the model is required to predict whether the relation $r$ is valid between subject and object, i.e., $y_{r|s,o} \in \{0, 1\}$, where $\theta$ is the parameters of neural networks, $K$ is the number of relation classes, and $\mathcal{C}$ and $\mathcal{R}$ are pre-defined object and relationship category sets. The objective is:

$$\min_\theta -E_i[y_{r|s,o}^i \log(\hat{y}_{r|s,o}^i) + (1 - y_{r|s,o}^i) \log(1 - \hat{y}_{r|s,o}^i)], \text{where } \hat{y}_{r|s,o}^i = f_\theta^{r^i}(I^i, b_s^i, b_o^i, c_s^i, c_o^i).$$

### 3.2 A REFINED SPATIAL RELATION BENCHMARK

We aim to benchmark various approaches for precise and physically grounded SRP. The two existing benchmarks most closely aligned to our problem are SpatialSense (Yang et al., 2019) and Rel3D (Goyal et al., 2020). SpatialSense is a realistic dataset annotated from Flickr (Plummer et al., 2017) and NYU (Nathan Silberman & Fergus, 2012) images, with various scenes including indoor, street and wilderness and many different objects such as humans, animals, plants, household items, etc. But because the precise definition of each relation category is not provided during the annotation process, it still inherits the linguistic biases of crowd annotators.

In Figure 2, we showcase some key issues in the original SpatialSense dataset: **(1)** mixing object-centric and viewer-centric relations together, which might have totally opposite meanings, e.g., from the perspective of the man in Figure 2 (a), the bass is on his left, but from the camera's view, the bass is to his right; **(2)** ambiguity caused by polysemous words, e.g., the "on"s in "books on shelf" and "cup on table" are different relationships; **(3)** language bias introduced by some idiomatic expressions, e.g., "boy in snow" but not "boy on snow". Rel3D aims to fix these issues but does so by

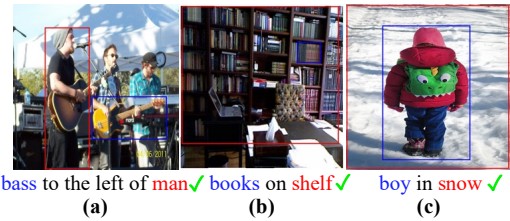

bass to the left of man✓ books on shelf✓ boy in snow ✓
(a)     (b)    (c)

Figure 2: Three issues of original SpatialSense: a) mixing the object-centric and viewer-centric annotations; b) ambiguity caused by polysemous words; c) language bias.

rendering a synthetic dataset completely in Blender (Community, 2018). Though Rel3D has diverse spatial layouts of objects, it is not photo-realistic – the background and object textures are simple, shown in Figure 1. While we evaluate our approaches on Rel3D as one of our two benchmarks for easy comparison with prior works, we seek a more thorough study of real-world in-the-wild images with diverse objects and scenes, closer to the domain of SpatialSense.

Therefore, we propose to construct a precise and physically grounded real-world dataset by relabeling SpatialSense. We propose precise, unambiguous, physically grounded meanings for various commonly used spatial relations for SpatialSense. For example, we restrict the meaning of "in" to be: "The subject is inside

Table 1: Statistics of Rel3D and SpatialSense+ in our benchmark.

| Dataset | #Predicate | #Train | #Validation | #Test |
|---|---|---|---|---|
| Rel3D | 30 | 20454 | 2138 | 4744 |
| SpatialSense+ | 9 | 5346 | 808 | 1100 |

the object, and the object must at least semi-enclose the subject." So, the "boy in snow" sample in Figure 2 (c) must be annotated as False. The full definitions of all the relation categories are shown in Appendix A.1. We randomly sample a subset of SpatialSense (amounting to 7254 relation triplets and 4418 images) and strictly relabel it by ourselves according to the new definitions. Eventually, we get a smaller but much cleaner version "SpatialSense+", that fixes the above-noted inconsistencies, including all the examples in Figure 2. Some basic statistics are shown in Table 1.

## 4 IN SEARCH OF AN ARCHITECTURE FOR SPATIAL RELATION PREDICTION

Recently, transformer architectures have achieved great success in various fields such as NLP (Devlin et al., 2019) and computer vision (Dosovitskiy et al., 2021), by taking advantage of the attention mechanism (Vaswani et al., 2017). Besides the advantages such as long horizon memory, attention in transformers explicitly models the pair-wise relationship between tokens, which is conceptually in line with our spatial relation prediction (**SRP**) task. Motivated by this, we aim to systematically study how to utilize the transformer architecture to extract spatial relation information from the images. We first introduce the basic components to solve this task and the architecture design principles for these steps. Based on these principles, we design multiple architectures for SRP. We introduce and compare the four most representative and natural designs in the main paper, while the remaining ones are presented in Appendix B.1.

### 4.1 DESIGN AXES

As introduced in Section 3.1, in the SRP task, the inputs are an image $I$ and two instance queries (subject and object) specified with their bounding boxes $b_s, b_o$ and object categories $c_s, c_o$. Because we focus on extracting relation features from visual representations, we omit the object categories from the inputs. In general, predicting the spatial relation between two objects from an image is composed of the following four components.

**Feature extraction:** Learning semantic representations from high-dimensional images is the first step of spatial relation prediction. In general, we produce visual features by a backbone such as CNNs (e.g., ResNet (He et al., 2016) or VGG (Simonyan & Zisserman, 2015)) or ViT (Dosovitskiy et al., 2021) which is representative of all the various transformer-based architectures in computer vision (Liu et al., 2021; Touvron et al., 2021b; Hassani et al., 2021; Chu et al., 2021).

**Query localization**: Given two bounding box queries, how to localize the queries and get their local features? One option is to use RoI Align, popular in object detection tasks, to localize the bounding

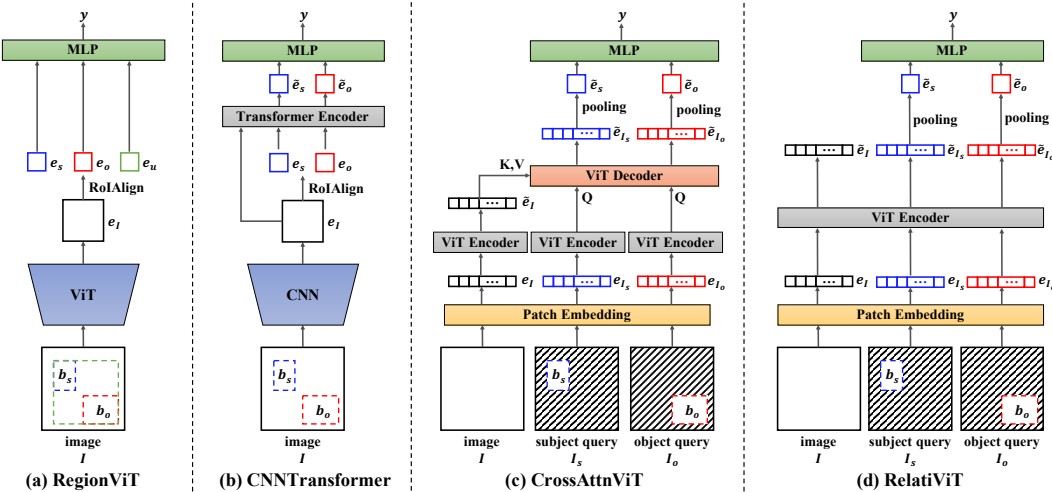

Figure 3: Four different architecture designs for SRP.

boxes in the low-resolution feature space (He et al., 2017). Another option is to localize the regions in the image space according to the bounding box coordinates, then mask out other regions, and get the local features by feeding the masked image through the feature extraction backbone.

**Context aggregation**: Next, how should we represent and use context information around the two query objects? For example, information about the obstacles between them is helpful to determine whether the relation is "next to". And the shadows might indicate some depth and contact information. Existing spatial relation prediction methods extract visual features of the union region of subject and object by RoI Align as context information (Li et al., 2017; Zhang et al., 2017b; Zellers et al., 2018). For transformer, the attention operation is natural to aggregate contextual features. Depending on where to input tokens of masked images $e_{I_s}, e_{I_o}$, the context aggregation in transformer can be divided into two types: 1) self attention: feed queries together with image patches as one sequence into the ViT encoder and then aggregate the context information in original image patches by self-attention layers; 2) cross attention: use queries as the input, then image features from the encoder as keys and values, and then use cross-attention layers to fuse the context information.

**Pair interaction**: Different from object recognition, our task is to recognize pair-wise relationship between subject and object, which cannot be learned by these two features separately. Therefore, modeling the interaction between two features is also critical for our task. The easiest way to combine the pair of features is concatenating them and feeding them into an MLP. Or in a more transformer-centric design, we could explicitly model such pairwise interactions through attention.

## 4.2 ARCHITECTURE DESIGNS

Based on the potential design options of each component discussed in Section 4.1, we make different combinations of these options and obtain various transformer-based architectures to extract spatial relationships. As shown in Figure 3, we present the four most representative designs here: (a) RegionViT, (b) CNNTransformer, (c) CrossAttnViT, (d) RelatiViT. Table 2 outlines their different choices for each design axis. And we compare more different designs in Appendix B.1.

**RegionViT:** Motivated by the object detection models (He et al., 2017; Li et al., 2021b), we design a straight-forward architecture to encode the visual features by a ViT, utilize RoI Align to localize subject and object embeddings $e_s, e_o$ efficiently from the low-resolution feature map $e_I$. To get the contextual information, we again use RoI Align to extract the feature $e_u$ in the union region of two bounding boxes, which is very common in the visual relation detection literature(Li et al., 2017; Zhang et al., 2017b; Zellers et al., 2018; Tang et al., 2019). And the pair-wise interaction is simply concatenating the features of subject and object. This architecture aims to study whether such a commonly used feed-forward style is suitable for ViT and the SRP task.

**CNNTransformer:** In Table 3 (b), different from RegionViT, we move the transformer from the feature extraction module to prediction head, i.e., we obtain the global visual embedding $e_I$ through a conventional CNN (e.g., ResNet (He et al., 2016) or VGG (Simonyan & Zisserman, 2015)), localize RoI embeddings $e_s, e_o$ by RoI Align, tokenize $e_I, e_s, e_o$ and then input them into a transformer to aggregate contextual and pair-wise information simultaneously. The purpose of this design is to investigate capability of transformers to capture relations on top of semantic representations.

Table 2: The choices of each component in different architectures.

| Architecture | Feature extraction | Query localization | Context aggregation | Pair interaction |
|---|---|---|---|---|
| RegionViT | ViT | RoI Align | RoI Align on union | Concatenation |
| CNNTransformer | CNN | RoI Align | Self attention | Self attention |
| CrossAttnViT | ViT | Masked image | Cross attention | Self attention |
| RelatiViT | ViT | Masked image | Self attention | Self attention |

**CrossAttnViT:** Motivated by the original transformer encoder-decoder architecture for machine translation tasks (Vaswani et al., 2017), we propose to attach a cross-attention decoder after ViT, which is shown in Figure 3 (c). In this pipeline, we first obtain the subject, and object query images $I_s, I_o$ by masking out the regions other than RoI, which completes the query localization step. We tokenize the original, subject and object query images by a patch embedding layer (Dosovitskiy et al., 2021). Patch embeddings of these three images are fed into a shared ViT encoder to extract the visual embeddings. Then the encoded embeddings of the original image and two masked images are put into the ViT decoder. In each layer of the decoder, there are one cross-attention layer and one self-attention layer (Vaswani et al., 2017). The cross-attention layer takes the embeddings of two masked images as queries, and the ones of the context image $\tilde{e}_I$ as keys and values, which aggregates contextual information. And the self-attention layer enables the message passing between subject and object. Finally, we take the output of ViT decoder $\tilde{e}_{I_s}, \tilde{e}_{I_o}$ and get the relation-grounded embedding $\tilde{e}_s, \tilde{e}_o$ through a pooling layer. This architecture is to study whether such an encoder-decoder structure commonly used for transformer-based models can effectively extract relation information.

**RelatiViT:** Besides the modular architectures mentioned above, we propose an end-to-end architecture RelatiViT, shown in Figure 3 (d). We utilize the same strategy as CrossAttnViT to localize the subject and object queries by image masking. After getting the embeddings $e_I, e_{I_s}, e_{I_o}$ from the patch embedding layer, we concatenate them as a long sequence and then feed them into the ViT encoder. During the forward process of ViT encoder, the self-attention mechanism completes context aggregation and pair interaction simultaneously. For example, subject tokens $e_{I_s}$ are able to attend to context features $e_I$ and object features $e_{I_o}$ because they are in the same sequence. After aggregating the contextual and pair-wise features in ViT encoder, we get $\tilde{e}_{I_s}, \tilde{e}_{I_o}$. We downsample these two embeddings by a pooling layer and get final relation-grounded embeddings $e_s, e_o$ for the subject and object respectively.

## 5 EXPERIMENTS

We evaluate the architectures proposed in Section 4.2 on our benchmark (Section 3.2), aiming to identify useful design principles. Then, we compare our best model with the existing methods. We also conduct a series of analytical experiments to probe our methods' performance.

### 5.1 EXPERIMENTAL SETUP

We train all the models on Rel3D and SpatialSense+ datasets. The input of each model includes an RGB image, bounding boxes and categories of the subject and object, and a spatial relation predicate between them, where all images are $224 \times 224$ and object categories are encoded into 300-dimensional embeddings by Glove (Pennington et al., 2014). Note that some methods may use part of the inputs, e.g., our methods proposed in Section 4.2 do not use the object categories. For fair comparisons, we adopt ViT-Base and ResNet101 as our ViT and CNN backbones which have close numbers of parameters. And we load the IBOT pretrained model (Zhou et al., 2022) for ViT-Base and ImageNet supervised pretrained model for ResNet101. During training, we use data augmentations on RGB images, e.g., random crop resize and color jitter. More training details are shown in Appendix A.2.

We report two evaluation metrics: 1) %Avg. Acc.: the average accuracy over relation classes, which avoids the overall accuracy being dominated by the relation classes with more samples; 2) %F1: the commonly-used F1 score in binary classification. We tune the hyperparameters on the validation set and the details about hyperparameters are listed in Appendix A.3. The checkpoints with the highest %Avg. Acc. on the validation set are taken for testing. We repeat the experiments 5 times with different seeds, and report the average values and standard deviations in Table 3 4 6 5.

### 5.2 BASELINES

**Naive baselines:** Following Goyal et al. (2020); Yang et al. (2019), we use two naive baselines: bbox-only and bbox-language to demonstrate whether other methods successfully utilize the visual information. The bbox-only method only takes bounding box coordinates and relation predicates

Table 3: Comparison between different designs. RelatiViT shows superior performance over RegionViT, CNNTransformer, and CrossAttnViT across two datasets.

| method | Rel3D | | SpatialSense+ | |
|---|---|---|---|---|
| | %Avg. Acc. | %F1 | %Avg. Acc. | %F1 |
| RegionViT | $72.43 \pm 0.55$ | $74.64 \pm 0.82$ | $76.20 \pm 1.04$ | $75.15 \pm 0.96$ |
| CNNTransformer | $71.13 \pm 0.23$ | $73.85 \pm 0.29$ | $68.21 \pm 0.92$ | $66.83 \pm 0.72$ |
| CrossAttnViT | $76.55 \pm 0.28$ | $78.76 \pm 0.55$ | $73.41 \pm 2.00$ | $71.77 \pm 3.40$ |
| RelatiViT | $\mathbf{80.09 \pm 0.33}$ | $\mathbf{82.05 \pm 0.49}$ | $\mathbf{80.02 \pm 0.73}$ | $\mathbf{78.32 \pm 1.01}$ |

as input, while bbox-language takes ground truth object categories in addition. Both two models are simple MLPs. Note that these are notoriously hard baselines to beat in the most relevant prior works (Yang et al., 2019; Goyal et al., 2020).

**Existing methods.** Because there are still no good solutions for SRP, we follow Goyal et al. (2020); Yang et al. (2019) to adapt five visual relation detection models (DRNet (Dai et al., 2017), Vip-CNN (Li et al., 2017), VTransE (Zhang et al., 2017a), PPR-FCN (Zhang et al., 2017b), Motifs-Net (Zellers et al., 2018), and RUNet (Lin et al., 2022)) to our task. Specifically, we replace the object detection module with the ground-truth bounding box coordinates and object classes, and then we modify the model output from K-way classification probabilities into a K-way binary classification probability conditioned on the input relation predicate. These baselines are divided into two groups based on the information used in the final MLP classifier: 1) vision: Vip-CNN and PPR-FCN, only taking vision features to make predictions; 2) vision+bbox+class: DRNet, VTransE, MotifsNet and RUNet, using visual features, bounding box and object class embeddings.

**Vision Language Models.** Beyond these spatial-relation-specific baselines, we also compare with four Vision Language Models (VLMs): MiniGPT-4 (Chen et al., 2023), LLaVA (Liu et al., 2023a), Gemini (Team et al., 2023) and GPT-4V (Achiam et al., 2023) to demonstrate the capability of these SOTA large models to capture spatial relations. We only evaluate these VLMs on the realistic dataset SpatialSense+. The specific version and input prompts are shown in Appendix.

## 5.3 COMPARISON BETWEEN DIFFERENT DESIGNS

As shown in Table 3, we compare the selected four architectures introduced in Section 4.2 on Rel3D and SpatialSense+. More fine-grained comparisons between more architectures by changing design options one by one are shown in Appendix B.1. By comparing their performance, we can obtain some insights about how to design transformer-based models for relation-based tasks.

CNNTransformer performs worse on both Rel3D and SpatialSense+ than RegionViT. This indicates that ViT is a stronger backbone for extracting visual relation information compared to CNN. An external transformer head is insufficient to compensate for the weak feature extraction performed by CNN. The better performance of "ViT + RoiAlign + Self-attention" than CNNTransformer in Appendix B.1 further verifies this argument by only replacing CNN with ViT.

Considering the results of both Rel3D and SpatialSense+, CrossAttnViT performs similarly to RegionViT. We further ablate the effect of "RoiAlign v.s. MaskImage" and "Self-attention v.s. Cross-attention" based on CrossAttnViT in Appendix B.1, but their performance also does not have significant differences. The commonly-used RoI Align technique and advanced transformer decoder do not benefit SRP performance of these cascading architectures, indicating that these cascading architectures are not effective in extracting relational information from the pretrained ViT.

RelatiViT performs significantly better than the other three architectures. In contrast to the three cascading models, RelatiViT feeds the patch embeddings $e_I, e_{I_s}, e_{I_o}$ as a unified sequence into the ViT encoder. Consequently, the long-horizon memory of the transformer allows the queries $e_{I_s}, e_{I_o}$ to refer to one another and attend to the contextual information in $e_I$ automatically, enabling simultaneous completion of query localization, context aggregation, and pair interaction. RelatiViT overcomes the limitations of other designs: 1) it employs a stronger ViT backbone than CNN, facilitating the extraction of relation-related features; 2) it does not introduce extra parameter overheads, mitigating optimization challenges; 3) the end-to-end modeling enables the extraction of relation-grounded information across all ViT layers.

## 5.4 COMPARISON WITH BASELINES

As shown in Table 4, to verify that transformer is conceptually suitable for spatial relation prediction tasks, we compare our best transformer-based model RelatiViT with the existing methods.

Table 4: Comparison with baselines. RelatiViT significantly outperforms all the existing methods.

| Method | Signal | Rel3D | | SpatialSense+ | |
|---|---|---|---|---|---|
| | | %Avg. Acc. | %F1 | %Avg. Acc. | %F1 |
| bbox-only | bbox | $74.35 \pm 0.77$ | $77.81 \pm 0.44$ | $77.55 \pm 0.78$ | $75.30 \pm 1.35$ |
| bbox-language | bbox+class | $73.11 \pm 0.29$ | $76.63 \pm 0.49$ | $73.88 \pm 1.39$ | $71.84 \pm 1.79$ |
| DRNet (Dai et al., 2017) | vision+bbox+class | $72.63 \pm 0.36$ | $75.28 \pm 0.49$ | $76.96 \pm 1.53$ | $75.74 \pm 1.44$ |
| VTransE (Zhang et al., 2017a) | vision+bbox+class | $72.50 \pm 0.50$ | $75.54 \pm 0.48$ | $77.10 \pm 0.84$ | $75.58 \pm 1.13$ |
| MotifsNet (Zellers et al., 2018) | vision+bbox+class | $72.42 \pm 1.34$ | $74.67 \pm 1.42$ | $75.58 \pm 1.18$ | $74.42 \pm 1.05$ |
| RUNet (Lin et al., 2022) | vision+bbox+class | $75.03 \pm 0.48$ | $77.83 \pm 0.41$ | $75.81 \pm 0.90$ | $74.20 \pm 0.73$ |
| Vip-CNN (Li et al., 2017) | vision | $69.72 \pm 0.67$ | $73.07 \pm 0.81$ | $69.71 \pm 1.33$ | $67.73 \pm 1.66$ |
| PPR-FCN (Zhang et al., 2017b) | vision | $71.20 \pm 1.42$ | $72.95 \pm 1.28$ | $70.68 \pm 0.97$ | $69.88 \pm 1.36$ |
| **RelatiViT (ours)** | vision | $\mathbf{80.09 \pm 0.33}$ | $\mathbf{82.05 \pm 0.49}$ | $\mathbf{80.02 \pm 0.73}$ | $\mathbf{78.32 \pm 1.01}$ |
| MiniGPT-4 (Chen et al., 2023) | vision+bbox+class | - | - | 50.00 | 33.33 |
| LLaVA (Liu et al., 2023a) | vision+bbox+class | - | - | 54.20 | 53.66 |
| Gemini Team et al. (2023) | vision+bbox+class | - | - | 59.17 | 59.55 |
| GPT-4V (Achiam et al., 2023) | vision+bbox+class | - | - | 68.26 | 66.51 |

With access to object classes, bbox-language cannot beat bbox-only, indicating that the object category information does not benefit this physically-grounded spatial relation task. Bbox-language does not obtain more useful information than bbox-only, but instead causes optimization difficulty due to the high-dimensional word embedding input, resulting in higher variance and worse performance.

We can see that none of the existing methods performs better than bbox-only on both datasets. Especially, the methods in "vision+bbox+class" group (DRNet, VtransE, Motifsnet and RUNet) perform slightly worse than bbox-only. We hypothesize that this is caused by shortcut learning (Geirhos et al., 2020; Wen et al., 2022) – deep neural networks prefer solutions with simpler decision rules rather than the complex intended ones. They are prone to simply predict the relationships based on the bounding box coordinates rather than learn the intended solutions by referring to the RGB input, which is further investigated in Appendix B.5. As for the methods in "vision" group (Vip-CNN and PPR-FCN), they perform poorly because it is difficult for CNN backbones to extract relation-grounded representations.

Furthermore, we extend our analysis to incorporate comparisons with advanced Vision Language Models (VLMs), showing the trend: GPT-4V > Gemini > LLaVA > MiniGPT-4. Despite their remarkable performance in VQA tasks, these large-scale pretrained models exhibit poor performance on the SpatialSense+ dataset. This discrepancy underscores the significance of our spatial relation prediction task as a pivotal benchmark for visual reasoning, essential for evaluating the capabilities of large-scale multi-modal models in the future.

Our best model RelatiViT outperforms all the baselines. With ViT's powerful representation capabilities and our end-to-end modeling of context aggregation and pair-wise interaction, RelatiViT successfully extracts the spatial relations from images and becomes the first one so far to clearly outperform the naive bbox-only baseline. For prediction visualizations, please refer to Appendix B.6.

Table 5: Ablation studies on the design components.

| PAIR INTERACTION | CONTEXT AGGREGATION | FEATURE EXTRACTION | REL3D | | SPATIALSENSE+ | |
|---|---|---|---|---|---|---|
| | | | %AVG. ACC. | %F1 | %AVG. ACC. | %F1 |
| ✔ | ✔ | ✔ | $80.09 \pm 0.33$ | $82.05 \pm 0.49$ | $80.02 \pm 0.73$ | $78.32 \pm 1.01$ |
| ✗ | ✔ | ✔ | $79.18 \pm 0.61$ | $81.30 \pm 0.57$ | $76.37 \pm 0.86$ | $75.01 \pm 0.87$ |
| ✗ | ✗ | ✔ | $70.08 \pm 0.41$ | $72.70 \pm 0.32$ | $74.04 \pm 0.75$ | $71.25 \pm 0.98$ |
| ✗ | ✗ | ✗ | $55.51 \pm 0.48$ | $57.59 \pm 1.07$ | $60.80 \pm 0.61$ | $59.34 \pm 0.58$ |

## 5.5 ANALYSIS ON RELATIVIT

We now study the winning RelatiViT architecture more closely through careful analyses.

**Ablation on design axes.** We perform ablation studies on the three components of RelatiViT: feature extraction, context aggregation, and pair interaction. Specifically, we (1) remove the ViT encoder, (2) add attention masks to context image tokens, and (3) corresponding query tokens, respectively. It is worth noting that we exclude query localization because it is an integral component. In Table 5, all three components are essential for the performance of RelatiViT. Feature extraction serves as the foundational module for extracting features from raw pixels, and its absence results in the most significant performance decline. Context aggregation plays a significant role in the spatial relation prediction task since the model relies on context information to deduce the relationship between the subject and object. The removal of pair interaction has a relatively smaller impact on performance.

Table 6: Accuracy of each class on SpatialSense+.

| Method | above | behind | in | right | on | under | next | front | left |
|---|---|---|---|---|---|---|---|---|---|
| bbox-only | **81.6 ± 2.4** | 71.3 ± 1.7 | 71.0 ± 3.6 | 93.5 ± 0.7 | **74.5 ± 2.8** | 76.5 ± 1.8 | 65.8 ± 2.6 | 71.0 ± 1.5 | 91.6 ± 0.9 |
| DRNet | 74.2 ± 4.0 | 65.9 ± 3.0 | **76.5 ± 5.3** | 93.4 ± 1.3 | 71.5 ± 3.2 | 73.6 ± 3.1 | **69.7 ± 2.5** | 75.7 ± 2.4 | 92.0 ± 4.3 |
| RelatiViT | 80.2 ± 1.3 | **78.1 ± 2.3** | 75.4 ± 3.8 | 93.5 ± 1.4 | 76.3 ± 1.5 | **79.7 ± 2.7** | 64.5 ± 2.9 | **79.8 ± 2.8** | 92.4 ± 1.3 |

**Performance on each class.** As shown in Table 6, we report the accuracy of each class on SpatialSense+. We compare three models, bbox-only, DRNet and RelatiViT, the most representative ones in naive baseline, existing methods and our architectures respectively. We can see that our method outperforms the bbox-only baseline on "above", "behind", "in", "on", "under" and "in front of", which requires visual information about depth, object poses and shapes, etc. Bbox-only performs better on "to the right of" than RelatiViT because according to the relation definition proposed in Section 3.2, this relation can indeed be predicted only by bounding box coordinates. The results on Rel3D are shown in Appendix B.2, and the conclusion is consistent. In summary, our RelatiViT successfully extracts effective visual representations for SRP.

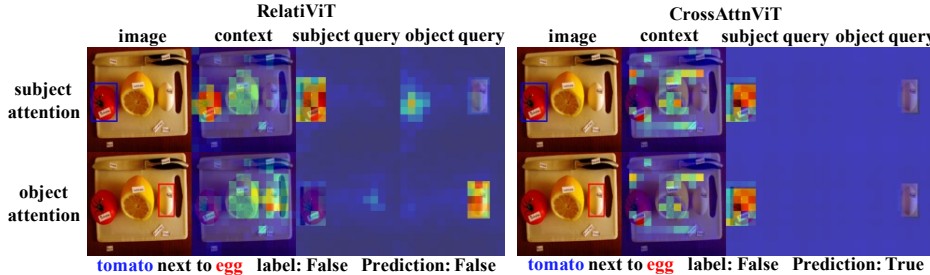

Figure 4: Comparison of attention maps. In each subfigure, the 1st and 2nd rows visualize the attention matrices averaged from the subject and object query embeddings respectively. The 1st column is the original image with the bounding box; the 2nd one is the attention map reflecting how much the query attends to the context image (context aggregation); the 3rd and 4th ones show the attention between two queries (pair-wise interaction).

**Attention map visualization.** To study how our approach refers to visual information, we visualize the attention maps of RelatiViT and CrossAttnViT, which reflect how much and in which area each query attends to the context image (context aggregation), itself and another query (pair-wise interaction). As shown in Figure 4, the model is required to recognize whether the tomato is next to the egg. We can see that besides focusing on the queries' regions, both subject and object queries of RelatiViT also attend to the lemon in between. Consequently, RelatiViT knows there is an obstacle between the tomato and egg, so it predicts "False". On the contrary, attention maps of CrossAttnViT are messy and hard to explain. This illustrates RelatiViT outperforms other methods by correctly attending to the critical regions of the context image and two masked query images. More visualizations are shown in Appendix B.7.

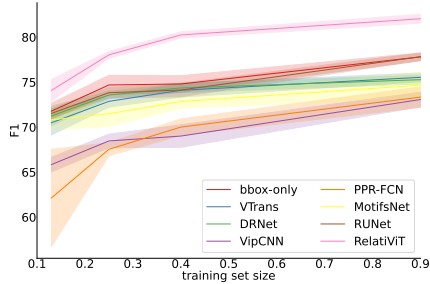

Figure 5: The performance on different sizes of the Rel3D training set.

**Effect of dataset size.** In Figure 5, we plot the F1 scores of each model with training set size varying in $[13\%, 25\%, 40\%, 90\%]$. The results show that RelatiViT outperforms all the baselines on all the training set sizes. Moreover, its performance on the 25% dataset is comparable with the best result of baselines on 90% training data. This verifies that our RelatiViT is more sample efficient by taking advantage of the inductive bias of pair-wise relation and prior knowledge in pretrained models.

## 6 CONCLUSION

In this paper, we benchmark the various transformer-based models to investigate the optimal way to predict precise and physically grounded relationships, a basic vision task that remains surprisingly challenging even with modern approaches, and which is essential for robotics manipulation and general scene understanding. Our experiments yield key insights about design principles for Transformer-based systems for this task, and identify a clear winner RelatiViT. This is the first system thus far to surpass naive baselines for this basic visual capability. And it will serve as an important starting point and baseline for future efforts in this direction.

## 7 REPRODUCIBILITY STATEMENT

The main implementations of our proposed models are in Section 4.2. The settings of the experiments and training details, e.g., data preprocessing and hyper-parameters, are in Section 5.1, Appendix A.2, and Appendix A.3.

## 8 ACKNOWLEDGEMENT

We are grateful to Chen Wang, Kaihua Tang, Long Chen, and Jiacheng You for fruitful discussions and comments.

This work is supported by the Ministry of Science and Technology of the People's Republic of China, the 2030 Innovation Megaprojects "Program on New Generation Artificial Intelligence" (Grant No. 2021AAA0150000), and the National Key R&D Program of China (2022ZD0161700). This work is also supported by NSF CAREER Award 2239301.

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

# A MORE DETAILS OF DATASETS AND EXPERIMENT SETUP

## A.1 FULL DEFINITION OF THE RELATION CLASSES

Table 7: The precise definition of each relation category in SpatialSense+.

| Relation | Definition |
|---|---|
| above | From the direction of gravity, the position of the subject is higher than object, the two are not in contact, and the two overlap in the direction of gravity. |
| behind | The depth of subject is larger than the object. If two objects are in very different directions, then skip this sample. |
| in | The subject is inside the object, and the object must at least semi-enclose the subject. |
| in front of | The depth of the subject is smaller than the object. If two objects are in very different directions, skip this sample. |
| next to | Subject is near to object, and there is no other obstacle between them. |
| on | The subject is on top of object and the two are in contact. If the subject is attached to the side of the object, it is not considered as "on". If the subject is semi-enclosed by the object, then it is not considered as "on". |
| to the left of | The subject is to the left of object from the labeler's view. If the difference in depth or height between the two is too large, skip the sample. |
| to the right of | The subject is to the right of the object from the labeler's view. If the difference in depth or height between the two is too large, skip the sample. |
| under | From the direction of gravity, the position of the subject is higher than the object. |

Table 8: The hyperparameters of RelatiViT.

| hyper-parameters | Rel3D | SpatialSense+ |
|---|---|---|
| epochs | 200 | 100 |
| early stop | 20 | 20 |
| optimizer | adamw | adamw |
| learning rate | 1e-5 | 1e-5 |
| lr schedule | cosine | cosine |
| lr warm-up epoch | 5 | 5 |
| weight decay | 1e-4 | 1e-3 |
| layer decay | 0.75 | 0.75 |
| query embedding pooling | max | max |

## A.2 TRAINING DETAILS

**Input preprocessing:** The images of Rel3D are center cropped to $224 \times 224$ and the ones of SpatialSense+ are resized to $224 \times 224$. We perform RandomCropResize and ColorJitter augmentations on the input images to make the training samples more diverse. At the same time, the bounding box coordinates are changed accordingly. Then we normalize the images and feed them into the models. As for the object class inputs, we use Glove (Pennington et al., 2014) to encode the words into 300-dimensional embeddings. The bounding box input is a 4-dimensional vector representing the coordinates of the upper left and lower right corners.

**Training objective:** We use binary cross-entropy loss to train all the models. Following the settings in Goyal et al. (2020) and Yang et al. (2019), we use the weighted loss for Rel3D and not for SpatialSense+, where the weights are computed according to the frequency of the relation categories.

**Model details**: For the CNN models, we use ResNet101 as the feature extraction backbone, and load the ImageNet supervised pretrained weights. For ViTs, we take ViT-Base as the backbone, and load the IBOT (Zhou et al., 2022) pretrained model by default. In Appendix B.3, we also

try other pretrained models, such as DEIT (Touvron et al., 2021a), MOCO-v3 (Chen et al., 2021), DINO (Caron et al., 2021), MAE (He et al., 2022) and CLIP (Radford et al., 2021).

### A.3 HYPERPARAMETERS

The hyperparameters of our best model RelatiViT are shown in Table 8. The hyperparameters of other baselines follows the setting in Goyal et al. (2020) and Yang et al. (2019).

### A.4 DETAILS OF VISION LANGUAGE MODELS

The specific version of each vision language model is shown in Table 9. The input prompts are shown below, where {} denote the placeholders for the names of the subject, object, and predicate, and the corresponding bounding box coordinates:

---

"Here are the definitions of relation predicates.
Above: From the direction of gravity, the position of the subject is higher than object, the two are not in contact, and the two overlap in the direction of gravity.
Behind: The depth of subject is larger than the object.
In: The subject is inside the object, and the object must at least semi-enclose the subject.
In front of: The depth of the subject is smaller than the object.
Next to: Subject is near to object, and there is no other obstacle between them.
On: The subject is on top of object and the two are in contact.
To the left of: The subject is to the left of object from the labeler's view.
To the right of: The subject is to the right of the object from the labeler's view.
Under: From the direction of gravity, the position of the subject is lower than the object.

According to the definitions, tell me whether the statement that the {} in blue bounding box (whose upper left corner coordinate is [x,y]={} and lower right corner coordinate is (x,y)={}) is {} the {} in the red bounding box (whose upper left corner coordinate is [x,y]={} and lower right corner coordinate is (x,y)={}) is True. Just tell me True or False."

---

Table 9: Version of each Vision Language Models.

| Model | Version |
| --- | --- |
| MiniGPT-4 | V2 |
| LLaVA | LLaVA-1.5 13B |
| Gemini | Gemini-Pro-Vision API |
| GPT-4V | GPT-4V Azure API |

## B MORE EXPERIMENTAL RESULTS

### B.1 MORE ARCHITECTURE DESIGNS AND COMPARISON RESULTS.

In Table 10, we change the design options of the four components introduced in Section 4.1 one by one in order to deliberately show the effect of each ax.

We can see that "ViT + RoiAlign + self-attention" is better than CNNTransformer, indicating that ViT features are more suitable for the spatial relation prediction task than CNN.

The performance of "ViT + MaskImage + self-attention" and "ViT + RoiAlign + self-attention" is similar, illustrating that the commonly-used RoiAlign technique (He et al., 2017) does not bring more benefits beyond the simple MaskImage strategy. But the latter is more suitable for end-to-end transformer modeling because of its simplicity.

Comparing "ViT + MaskImage + self-attention" with CrossAttnViT, we find the self-attention relation decoder performs similarly to the cross-attention. We hypothesize the reason is that different

from the machine translation tasks where the cross-attention decoder is commonly used, the self-attention and cross-attention decoder are appropriately equivalent because the queries and keys are in the same domain in our spatial relation prediction task.

Finally, we compare the cascading architecture "ViT + MaskImage + self-attention" with RelatiViT, and the superior performance of RelatiViT illustrates the effectiveness of the non-cascading design.

Table 10: Fine-grained comparison between more designs.

| Model | Rel3D | | SpatialSense+ | |
|---|---|---|---|---|
| | %Avg. Acc. | %F1 | %Avg. Acc. | %F1 |
| CNN + RoiAlign + Self-attention (CNNTransformer) | $72.09 \pm 0.52$ | $74.86 \pm 0.35$ | $68.21 \pm 0.92$ | $66.83 \pm 0.72$ |
| ViT + RoiAlign + Self-attention | $75.95 \pm 0.70$ | $79.10 \pm 0.57$ | $72.42 \pm 1.28$ | $71.61 \pm 1.35$ |
| ViT + MaskImage + Self-attention | $74.37 \pm 1.25$ | $77.01 \pm 1.23$ | $74.40 \pm 2.14$ | $72.59 \pm 2.33$ |
| ViT + MaskImage + Cross-attention (CrossAttnViT) | $76.55 \pm 0.28$ | $78.76 \pm 0.55$ | $73.41 \pm 2.00$ | $71.77 \pm 3.40$ |
| ViT + MaskImage + Self-attention + End-to-End (RelatiViT) | $80.09 \pm 0.33$ | $82.05 \pm 0.49$ | $80.02 \pm 0.73$ | $78.32 \pm 1.01$ |

## B.2 ACCURACY ON EACH CLASS ON REL3D

The detailed results of accuracy on each relation class on Rel3D are shown Table 11. We can see that our RelatiViT outperforms the bbox-only and DRNet on most of the relation classes. Especially on classes where visual information is critical such as "lean against", "inside", "aligned to", "touching" and "on top of", etc., our method is significantly better than the baselines, indicating that our method successfully extracts the visual representations containing spatial relation information.

Table 11: Accuracy of each class on Rel3D with average values and standard deviations.

| Method | over | behind (wrt you) | to the side of | on | to the side of (wrt you) |
|---|---|---|---|---|---|
| bbox-only | $81.42 \pm 1.38$ | $93.91 \pm 1.15$ | $66.95 \pm 1.15$ | $82.93 \pm 1.94$ | $87.22 \pm 1.84$ |
| DRNet | $76.65 \pm 1.09$ | $\mathbf{95.78 \pm 1.06}$ | $64.25 \pm 3.52$ | $81.29 \pm 1.48$ | $85.28 \pm 1.67$ |
| RelatiViT(ours) | $\mathbf{84.68 \pm 1.68}$ | $94.06 \pm 1.17$ | $\mathbf{72.82 \pm 1.80}$ | $\mathbf{87.07 \pm 0.98}$ | $\mathbf{95.56 \pm 1.36}$ |

| method | to the right of | in front of | points towards | behind | near |
|---|---|---|---|---|---|
| bbox-only | $54.17 \pm 2.17$ | $62.03 \pm 4.35$ | $51.30 \pm 0.96$ | $59.09 \pm 2.54$ | $91.60 \pm 0.27$ |
| DRNet | $49.50 \pm 3.52$ | $58.28 \pm 2.78$ | $50.43 \pm 1.98$ | $57.27 \pm 3.51$ | $84.79 \pm 2.17$ |
| RelatiViT(ours) | $\mathbf{63.17 \pm 1.62}$ | $\mathbf{71.56 \pm 1.36}$ | $\mathbf{72.61 \pm 1.06}$ | $\mathbf{62.73 \pm 3.66}$ | $\mathbf{91.85 \pm 1.21}$ |

| method | points away | aligned to | faces towards | leaning against | faces away |
|---|---|---|---|---|---|
| bbox-only | $50.32 \pm 2.49$ | $61.21 \pm 2.31$ | $53.50 \pm 0.62$ | $70.00 \pm 2.16$ | $55.84 \pm 1.16$ |
| DRNet | $52.70 \pm 2.49$ | $64.14 \pm 2.70$ | $55.50 \pm 1.25$ | $68.65 \pm 1.58$ | $54.42 \pm 1.61$ |
| RelatiViT(ours) | $\mathbf{57.46 \pm 2.11}$ | $\mathbf{72.59 \pm 2.21}$ | $\mathbf{64.00 \pm 2.26}$ | $\mathbf{80.81 \pm 2.32}$ | $\mathbf{56.10 \pm 1.86}$ |

| method | in | in front of (wrt you) | far from | on top of | touching |
|---|---|---|---|---|---|
| bbox-only | $80.07 \pm 2.96$ | $95.00 \pm 1.11$ | $\mathbf{87.66 \pm 1.08}$ | $86.22 \pm 0.54$ | $67.27 \pm 3.75$ |
| DRNet | $74.41 \pm 4.42$ | $93.06 \pm 3.04$ | $73.62 \pm 1.56$ | $82.84 \pm 1.10$ | $67.12 \pm 2.07$ |
| RelatiViT(ours) | $\mathbf{85.72 \pm 1.23}$ | $\mathbf{96.39 \pm 1.42}$ | $80.85 \pm 2.77$ | $\mathbf{91.76 \pm 0.79}$ | $\mathbf{75.45 \pm 1.70}$ |

| method | covering | to the left of (wrt you) | to the right of (wrt you) | to the left of | below |
|---|---|---|---|---|---|
| bbox-only | $86.94 \pm 0.81$ | $94.60 \pm 0.64$ | $94.52 \pm 0.44$ | $63.33 \pm 7.15$ | $81.49 \pm 2.65$ |
| DRNet | $84.01 \pm 1.26$ | $92.06 \pm 0.56$ | $94.70 \pm 1.27$ | $62.00 \pm 7.77$ | $78.24 \pm 1.88$ |
| RelatiViT(ours) | $\mathbf{88.03 \pm 0.74}$ | $\mathbf{95.16 \pm 0.53}$ | $\mathbf{97.22 \pm 0.52}$ | $\mathbf{75.33 \pm 4.40}$ | $\mathbf{88.78 \pm 1.01}$ |

| method | under | passing through | inside | around | outside |
|---|---|---|---|---|---|
| bbox-only | $83.83 \pm 1.70$ | $\mathbf{73.58 \pm 1.19}$ | $71.33 \pm 1.35$ | $79.76 \pm 0.71$ | $63.33 \pm 4.86$ |
| DRNet | $83.19 \pm 1.04$ | $72.08 \pm 4.68$ | $71.50 \pm 2.76$ | $82.80 \pm 2.02$ | $68.33 \pm 3.33$ |
| RelatiViT(ours) | $\mathbf{90.21 \pm 0.80}$ | $72.08 \pm 0.96$ | $\mathbf{80.17 \pm 1.43}$ | $\mathbf{85.98 \pm 1.02}$ | $\mathbf{72.50 \pm 5.65}$ |

## B.3 ANALYSIS OF PRETRAINED MODELS

**Effect of pretrained representations.** We investigate the effect of pretraining methods for the spatial relation prediction task. We compare the performance of RelatiViT with different pretrained models: random initialization (scratch), DEIT (Touvron et al., 2021a), MOCO-v3 (Chen et al., 2021), CLIP (Radford et al., 2021), DINO (Caron et al., 2021), IBOT (Zhou et al., 2022) and

MAE (He et al., 2022). We evaluate these pretrained models on both Rel3D and SpatialSense+. Looking at the average performance over two settings (the third cluster of bars) in Figure 6, we can see that most of the self-supervised methods are better than the supervised one (DEIT) and training from scratch, except MOCO-v3. Among all the self-supervised learning on ImageNet, the self-distillation methods (IBOT and DINO) performs best. And IBOT performs better than DINO because the patch-wise pretraining of IBOT implicitly learns the relation information. Besides, CLIP performs well (only worse than IBOT) because it takes advantage of the multi-modal contrastive learning and the larger-scale pretraining dataset.

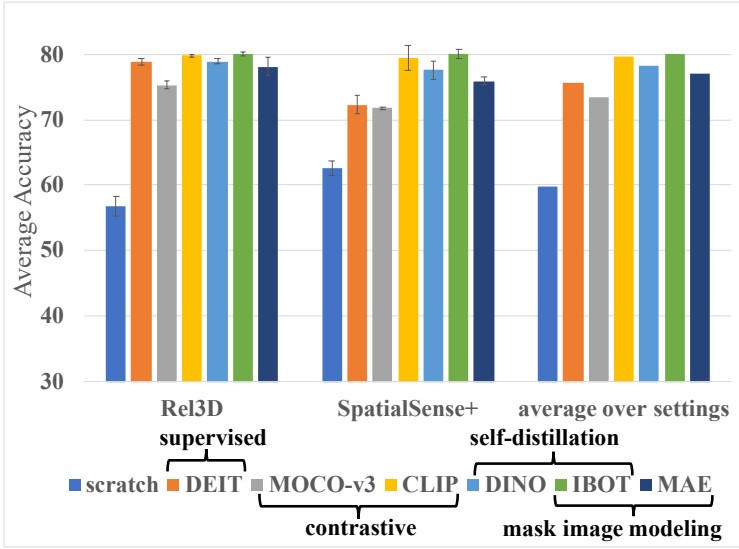

Figure 6: Effectiveness of different pretraining methods.

## B.4 COMPARISON WITH CNN MODELS WITH POSITIONAL EMBEDDINGS

There are two main differences between transformers and CNN: attention mechanism and positional embeddings. To further strengthen our argument that transformers leverage the attention mechanism to capture the relation information, we compare our RelatiViT to CNN-based models with positional embeddings implemented by CoordConv Liu et al. (2018). Specifically, we incorporate positional embeddings into the input tensor of the first layer of ResNet in DRNet, RUNet, and CNNTransformer. As shown in Table 12, the performance of these CNN-based models remains inferior compared to RelatiViT, indicating that they do not significantly benefit from positional embeddings. This outcome reinforces our hypothesis that RelatiViT's superior performance is largely due to its enhanced ability to capture spatial relationships through attention mechanisms, as opposed to the sliding convolution in traditional CNNs.

Table 12: Comparison with CNN-based models with CoordConv.

| Model | SpatialSense+ | |
| --- | --- | --- |
| | %Avg. Acc. | %F1 |
| DRNet + CoordConv | $76.86 \pm 0.96$ | $75.44 \pm 0.90$ |
| RUNet + CoordConv | $75.91 \pm 0.73$ | $74.68 \pm 1.26$ |
| CNNTransformer + CoordConv | $68.69 \pm 1.23$ | $66.47 \pm 1.81$ |
| RelatiViT | $80.02 \pm 0.73$ | $78.32 \pm 1.01$ |

## B.5 EFFECTS OF VISUAL FEATURES ON DIFFERENT MODELS

To quantitatively verify our statement that the existing methods tend to make predictions mainly according to bounding box coordinates rather than image content, we conduct an intervention experiment by replacing the input images with random ones while keeping the original bounding boxes

during evaluation. This is a common technique to diagnose the causal effect of each input signal (Wen et al., 2022; Chuang et al., 2022; Bruce et al., 2024). As shown in Table 13, we report the *prediction flipping ratio*, which measures the frequency of predictions changing from True to False, or vice versa, across all samples. A higher flipping ratio suggests greater reliance on image content, while a lower ratio indicates a tendency to base predictions on bounding box coordinates. It is evident that the prediction flipping ratios for CNNTransformer and RelatiViT are comparable to each other, and are notably higher than those for RUNet and DRNet. This result suggests two key insights:

1) Our transformer-centric design principles (including feature extraction, query localization, context aggregation, and pair interaction), which do not incorporate bounding box coordinates into the input unlike previous methods, are indeed more inclined to utilize visual information for predicting spatial relations. This finding supports the rationale behind our design choices, especially in the context of spatial relation prediction tasks.

2) Despite a similar emphasis on visual information, RelatiViT significantly outperforms CN-NTransformer. This indicates the superior capability of the vision transformer backbone in extracting spatial relation information compared to CNNs.

Table 13: Intervention results on SpatialSense+. The *prediction flipping ratio* denotes the percentage of samples whose prediction flips after we intervene their input images.

| Model | %Prediction flipping ratio |
|---|---|
| DRNet | $2.33 \pm 2.11$ |
| RUNet | $6.99 \pm 2.26$ |
| CNNTransformer | $30.18 \pm 1.69$ |
| RelatiViT | $29.47 \pm 1.88$ |

## B.6 VISUALIZATIONS OF PREDICTIONS

We show some examples of the output predictions of the baselines and RelatiViT in Figure 7. Our RelatiViT performs significantly better and more reasonably than all the baselines. The performance on these hard samples that cannot be handled by the naive methods such as bbox-only illustrates that ReltiViT predicts the spatial relations based on the visual information, e.g., object contact, occlusion, depth, background context, etc.

## B.7 MORE VISUALIZATIONS OF ATTENTION MAPS.

Table 8 shows more visualization results of the attention maps of RelatiViT and ViT cross-attention decoder. We can see that the attention maps of RelatiViT are clearer, and the subject and object queries attend to the contextual information between the two, in addition to referring to their own regions, which is conceptually reasonable for predicting spatial relations.

## B.8 RESULTS ON ORIGINAL SPATIALSENSE

We also conduct experiments on the original SpatialSense dataset and the results are shown in Table 14. Because of the imprecise definition of the relation classes, the original dataset still contains language biases during annotating. As a result, it is insufficient to benchmark the authors' intended objective – predicting spatial relations based on visual information. Consequently, our method RelatiViT performs similarly to the baselines. This illustrates the necessity to define the precise, unambiguous and physically grounded relation categories and relabel the SpatialSense dataset.

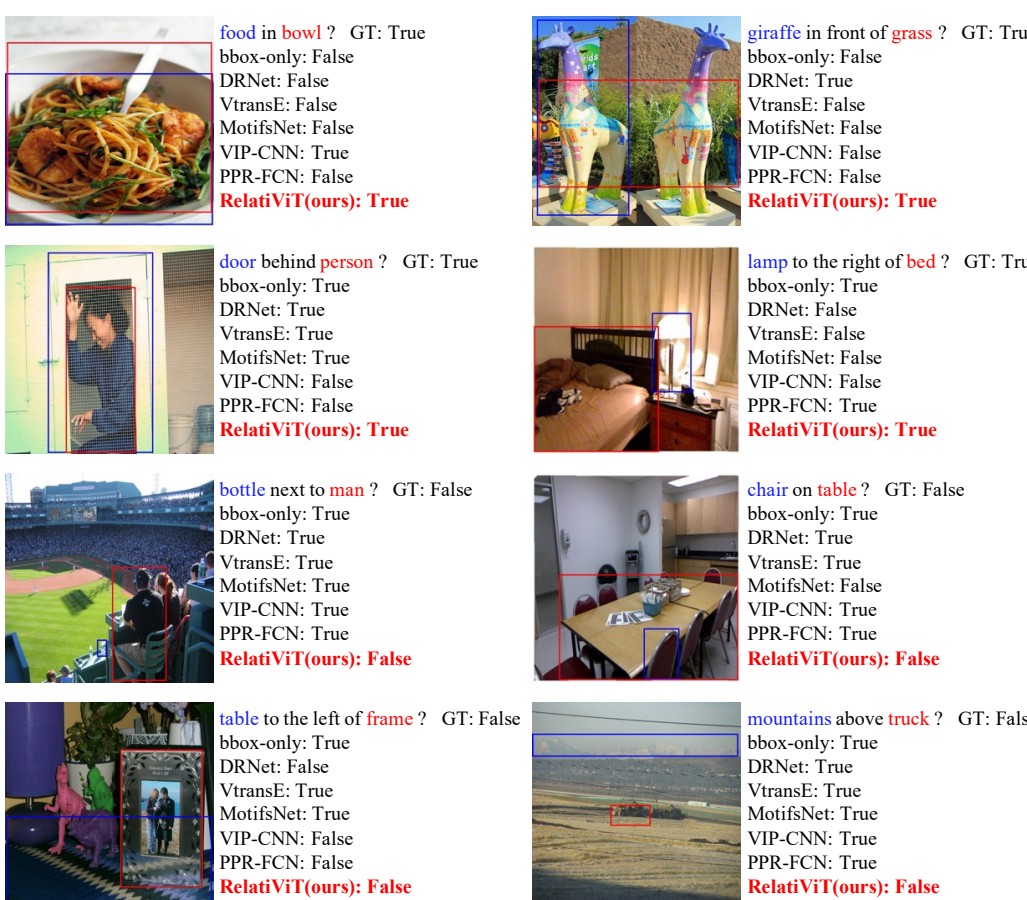

Figure 7: The prediction examples of the models, where GT denotes "ground truth".

Table 14: The results on the original SpatialSense.

| method | original SpatialSense | |
|---|---|---|
| | %Avg. Acc. | %F1 |
| bbox-only | $64.91 \pm 0.71$ | $66.06 \pm 1.29$ |
| bbox-language | $61.88 \pm 0.66$ | $63.04 \pm 1.05$ |
| DRNet (Dai et al., 2017) | $66.96 \pm 0.98$ | $69.32 \pm 1.25$ |
| VTransE (Zhang et al., 2017a) | $64.96 \pm 0.96$ | $66.42 \pm 1.27$ |
| MotifsNet (Zellers et al., 2018) | $65.21 \pm 1.04$ | $67.88 \pm 1.17$ |
| Vip-CNN (Li et al., 2017) | $63.29 \pm 0.47$ | $64.80 \pm 0.65$ |
| PPR-FCN (Zhang et al., 2017b) | $59.44 \pm 0.59$ | $62.33 \pm 0.56$ |
| RelatiViT (ours) | $65.79 \pm 1.12$ | $68.55 \pm 1.63$ |

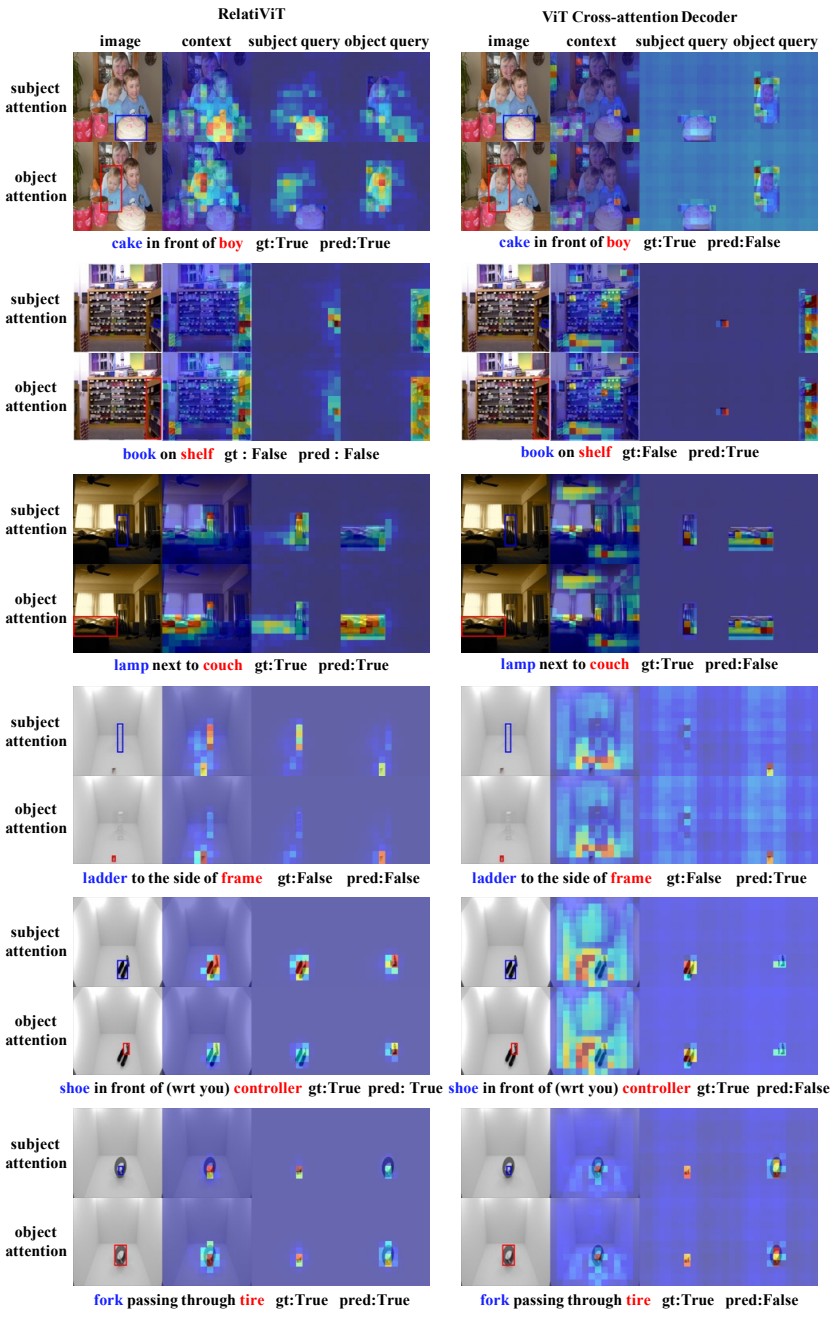

Figure 8: More attention map visualizations of End-to-end ViT and ViT cross-attention decoder on Rel3D and SpatialSense+, where "gt" denotes the ground truth label and "pred" is the prediction of the model.

