# OpenReview forum: "Can Transformers Capture Spatial Relations between Objects?"
_ICLR.cc/2024/Conference — ICLR 2024 poster_

### Official Review · Reviewer_ubwR · 2023-10-26

**Soundness:** 3 good
**Presentation:** 4 excellent
**Contribution:** 3 good
**Rating:** 8
**Confidence:** 4

**Summary:**

This paper studies the task of spatial relation prediction. The paper observes that prior works have focused on semantic relationships that may not be physically grounded and which can be ambiguous. To alleviate this issue, they define more precise and physically grounded definitions for relationships and reannotate an existing dataset (SpatialSense) to provide an improved test bed. Furthermore, the propose a new architectural choice that performs well on both existing synthetic datasets as well as the newly annotated benchmark.

**Strengths:**

- The paper is very well motivated with a clear explanation of how the proposed analysis and method fits within the larger scope of the field.
- The paper identifies a very interesting and nuanced ambiguity in existing annotations and provides clear examples of such issues.
- The paper provides a good explanation for the different subtasks that a model needs to do and the corresponding architectural components that do it in Sec 4.1.
- I appreciated that the authors included the very strong performing naive baselines and elaborated on why they perform well.
- The paper has a nice flow where the proposed architecture follows logically from the analysis and design considerations presented in the work.

**Weaknesses:**

I found three weaknesses in the paper: (1) it is unclear if the newer annotation provide a better evaluation or a simpler task; (2) ablations are not very clear; (3) several statements are not supported by the results. I have listed the major concerns below. I also included some minor concerns that should be dealt with as extra suggestions, and do not need to be addressed in the rebuttal.


- The paper very nicely motivates language ambiguity as an issue for obtaining spatial relations. However, it is unclear if the newer annotation scheme results in more accurate annotation or simply annotation that is better correlated with a relative location prior.
    - Comparing Tables 4 and 11, one finds that all methods achieve a higher absolute score on the newer dataset, while the bbox baselines and the proposed method seeing stronger gains. Furthermore, Table 7 shows that the new annotation seems more specific regarding annotations being done in the labelers point of view or gravity to minimize ambiguity. As a result, while it is possible that the new definitions are more precise and a better evaluation, it is also possible that they are more strongly correlated with the relative locations in the image, which would introduce a strong prior.
    - Some of the definitions in Table 7 support the point above. For example, "A in front of B" is defined as A is closer to the camera than B. While this is a less ambiguious definition, it is a much easier one than the typical usage of "in front of" which often also requires reasoning about the subject and object's relative orientation. For example, if two people are facing away from the camera, the further away person would be in front of the person closer to the camera. This is still unambiguous, but the prediction would be more difficult as it requires the model to both reason about relative depth as well as orientation of each person.
    - Said differently, the newer annotation scheme reduces the ambiguity but it also makes the relationships less dependant on context for cases like front/behind where the RelatiViT shows greatest performance improvements on the baseline (Table 6). This can be shown by the great improvements in performance of the bbox-only baseline. It would be nice to provide some qualitative analysis or quantitative comparisons that address this.

- The ablations are not very clear and the conclusions drawn do not seem supported by the results.
    - It is unclear how the method works without Pair Interaction, Context Aggregation, or Feature Extraction. For example, pair interaction is stated as the MLP that combines the two features, how is the prediction made without it? Additionally, how does one make predictions without feature extraction? is it based on pixel values, patch embeddings, something else? I think it is very important to explain exactly what each ablated model looks like.
    - The ablations are additive, so that the feature extraction ablation is applied to an already ablated model. While this is valid experiemntal choice, it makes it difficult to understand the most critical modules as they are not ablated independatly of each other. This is more important in this problem where we know that feature extract is not actually very important since bbox baselines perform very well, as shown in the paper.

- Several statements or claims are not fully supported/substantiated.
    - The statement `it is difficult for CNN backbones to extract relation-grounded representations` (sec 5.4) is strongly confounded by the use of positional embeddings. Specifically, based on the description, it seems that all methods (ViTs) have access to positional embeddings, while the CNN-Transformer baseline doesn't. While I understand that this is a common choice for CNNs, it seems like a strong confounder here due to the strong performance of bbox-only baseline. It is possible that ViTs outperform CNNs due to the explicit inclusion of positional embeddings rather than due to attention explicitly modeling pairwise relationships as suggested in Sec 4. While I think that attention is playing a strong role here, I think providing the CNN model with some positional embeddings, eg, through CoordConv (Liu et al, NeurIPS 2018), would help clairfy this point and explain whether CNNs are still poor feature extractors even if another model performs context aggregation afterwards (following the design axes in sec 4.1).
    - The paper states that `[prior methods] are prone to simply predict the relationships based on the bounding box coordinates rather than learn the intended solutions by referring to the RGB input.` (Sec 5.4) but that statement is not supported by the results. Most methods are consistently outperformed by the bbox-only baseline sugguestion that they are probably doing something other than predicting relationships baseline on bounding box coordinates. While I agree that those models are likely learning shortcuts, it seems odd that this shortcut would be using the bounding box coordinates if they are all underperforming the bounding-box baseline. Without conducting some further analysis, it is difficult to make such statements about what those models are doing.
    - The description under Table 6 doesn't match the results. The paper states `Bbox-only performs better on “to the right of” than RelatiViT because according to the relation definition proposed in Section 3.2` which is not true. All methods achieve the same mean performance on the `right` relationship, while bbox outperforms RelatiViT on `Next` and `Above`.

- I believe that there is a typo in the equation in Sec 3.1. I think you want to minimize the negative log likelihood, while the current term minimizes the log likelihood, as for $y{=}1$ the term would be minimized by estimating $\hat{y}$ to be 0. If I am wrong, could you please clarify why this is the desired objective?




------------
**Minor concerns and suggestions:** Those are just things that I thought would improve the paper or minor issues I noticed. You do not need to address any of them and I did not factor them into any ratings.

- While the authors do not have to fully explain every prior method used as a baseline, I think it would have been good to explain DRNet as it is used as the other baselines in the detailed comparisons presented in Table 6.
- I think the conclusion of `In summary, RelatiViT successfully extracts effective visual representations for SRP` (sec 5.5) doesn't seem well supported. While it is great that a method can outperform such baselines and I appreciate the authors making such baselines very prominent in the paper, the results further showcase that we still do not have any methods that perform well on this task as their performance is still very close to models that only get bounding boxes. I think the paper is a good step towards better modeling and evaluation, but it is unclear if we have any methods that extract good feature for SRP.
- Table 4: I think that adding some columns to clarify the difference between the methods would be useful. Speficially, Sec 4.1 did a nice job at clarifying the main modules for this task, and I thought that clarifying them in Table 4 would improve it.
- Figure 4: A little bit of white space or gap between the plots would make the figure a lot more easier to parse especially since the attention maps meld into each other.

**Questions:**

- Are the newer annotations simplifying the task or providing a stronger evaluation? I would appreciate it if the authors commented on the points raised in the weaknesses and provided some quantitative or qualitative results supporting it. I want to note that I think the paper is still valuable even if it was simplifying the annotation, it is just a matter of better explaining the contributions and limitations of the proposed dataset as an evaluation for this task.
- How are the ablations conducted? I would appreciate it if the authors could explain what the model looks like under each ablations. As noted above, I think it would be stronger if the ablations are conducted independantly. If this is not possible (due to some reliance on each other), it would be good to explain that.
- Do positional embeddings explain the difference between CNN and Transformers in Table 3?

---

> ### Author Response · Authors · 2023-11-21
>
> Thank you very much for the insightful suggestions and supportive comments. We hope the following response will address your concerns.
>
> **Q1:** It is unclear if the newer annotation provides a better evaluation or a simpler task. It is possible that the new annotations are more strongly correlated with the relative locations in the image, which would introduce a strong prior.
>
> **Answer:** More precise definition does indeed result in an easier task. Imprecise definitions and the resulting inconsistent annotations lead to "irreducible errors" and therefore worse performance metrics. More importantly, the resulting models are less useful, because it is not clear what meaning to assign to the models if the task is specified in this imprecise/inconsistent way.
>
> **Q2:** The newer annotation scheme reduces the ambiguity but it also makes the relationships less dependent on context. For example, the definition of “in front of” in the paper is a much easier one than the typical usage which often also requires reasoning about the subject and object's relative orientation.
>
> **Answer:** Thanks for this example. In our opinion, this is only true for objects with "a canonical front". Not all object categories have such properties -- most don't. Furthermore, note that this involves reasoning through semantics. We do believe there is potential value for such a task, but it is worthwhile to decompose the geometric and semantic components of the task. i.e., first use our geometric notion of spatial relationships, then identify the semantic spatial orientations of the objects, and then combine these inferences into a "semantic spatial relationship" such as "human A is in front of human B".
>
> **Q3:** It is unclear how the method works without Pair Interaction, Context Aggregation, or Feature Extraction.
>
> **Answer:** In our model, Pair Interaction is implemented by the attention mechanism between subject and object query tokens within the transformer. We ablate it by applying an attention mask on the attention weight between these two tokens. For Context Aggregation, similarly, we employ an attention mask on the attention weights between the query tokens and the image patch tokens. For Feature Extraction, we remove the ViT Encoder and relied solely on the patch embeddings of the image patches to predict spatial relations. We will describe the ablation experiment setup more clearly in our revision.
>
> **Q4:** Provide the CNN model with some positional embeddings, e.g., through CoordConv.
>
> **Answer:** Thanks for your valuable suggestion! Following your advice, we incorporate positional embeddings into the input tensor of the first layer of ResNet in both DRNet and RUNet. The results, as illustrated in the table below, indicating that the CNN-based models (DRNet and RUNet) did not benefit significantly from the addition of positional embeddings. Their performance remains inferior compared to RelatiViT. This outcome reinforces our hypothesis that RelatiViT’s superior performance is largely due to its enhanced ability to capture spatial relationships through attention mechanisms, as opposed to the sliding convolution used in traditional CNNs.
>
> |  Method   |  %Avg. Acc.  | %F1  |
> | :----: | :----: | :----: |
> | DRNet-CoordConv  | $76.86\pm0.96$ | $75.44\pm0.90$ |
> | RUNet-CoordConv  | $75.91\pm0.73$ | $74.68\pm1.26$ |
> | RelatiViT (ours) | $80.02\pm0.73$ | $78.32\pm1.01$ |
>
> **Q5:** More analysis is required to make the statement that the prior methods tend to make predictions mainly according to bounding box coordinates.
>
> **Answer:** Thank you very much for this suggestion. To verify whether the models predominantly rely on bounding box coordinates over image content, we conducted an intervention experiment by replacing the input images with random ones during testing. As shown in the following table, we report the *prediction flipping ratio*, which measures the frequency of predictions changing from True to False, or vice versa, across all samples. A higher flipping ratio suggests greater reliance on image content, while a lower ratio indicates a tendency to base predictions on bounding box coordinates. We find that DRNet and RUNet exhibit significantly lower flipping ratios than RelatiViT, indicating that RelatiViT places more emphasis on visual cues to when predicting spatial relations, contrary to the baselines focusing mainly on bounding box coordinates.
>
> | Method | %Prediction flipping ratio |
> | :----: | :----: |
> | DRNet | $2.33 \pm 2.11$ |
> | RUNET | $6.99 \pm 2.26$ |
> | RelatiViT | $29.47 \pm 1.88$ |
>
> **Q6:** Typo in loss function.
>
> **Answer:** Thank you very much for pointing this out. We will correct it in our revision.
>
> Once again, we are grateful for your precious time, insightful suggestions, and supportive comments. We will revise our paper according to your suggestions.

---

> ### Comment · Reviewer_ubwR · 2023-11-21
> **Reviewer Response**
>
> Thank you for engaging with all my comments and points. I respond below to each of those points. Overall, the points raised in Q1/2/6 are all addressed. I would appreciate the authors response regarding Q 3,4, and 5.
>
> Q1/Q2: The point made here is the newer annotations are more precise, and while they may make the task easier, this is primarily by resolving ambiguity (A1) and relying more on geometric rather than semantic relationships (A2). Overall, I see your point and mostly agree with it. One extra point I will add is that this is not just geometric, it's geometric relationships defined in a primarily ego-centric frame of reference. This changes things a bit as you can have well-defined relationships that do not depend on the observer, and such relationships would have much weaker relationships to the bounding box locations as they would be less view-dependent. One suggestion, in the spirit of your work, would be to revise the naming in Table 7 as many of those relationships are defined in an object-centric frame of reference; eg, in-front-of is currently defined with respect to the object, however, the relationship is a function of both the viewer and the object and is completely independent of the object's pose, just its location. Anyway, I think any remaining disagreements are more subjective, as a result, I think those points have been addressed.
>
> Q3: Thanks for the clarification. For pair interaction, does this mean that your attention matrix simply becomes a block matrix (with 4 blocks, 2 on the diagonal and 2 off the diagonal), where off-diagonal blocks are set to 0? if so, isn't that just equivalent to self attention with shared weights followed by concatenation?
>
> Q4: I appreciate the additional experiments, however, I find it difficult to make fair comparisons as both DRNet and RUNet have specialized architectures and hence, adding CoordConv to them does not provide a clear comparison to your method. In my review, I suggested the impact of positional embeddings specifically on the results in Table 3 as this, to me, is the key results table in the paper as it very nicely tests the claims of the paper. The question I have pertains to adding positional embeddings before the RoI align in Figure 3b. The reason I think this is an important experiment is because it provides the transformer following the CNN encoder with positional embeddings, something that it natively gets in the ViT encoders. Furthermore, by comparing the model to the adapted CNN-Transformer Figure 3b, we can control for other effects such as the architectural choices of previous approaches potentially being counterproductive.
>
> Q5: I appreciate going through the extra work. Overall, I really liked the experimental design of this experiment as it tests for sensitivity in a very nice way. However, my critiques in the above point hold. DRNet and RUNet do not provide a clear comparison to RelatiViT that addresses the main hypothesis/question of the paper as stated by the title and elaborated on in section 4.

---

> > ### Author Response · Authors · 2023-11-22
> >
> > Thank you very much for your feedback. We are pleased to know that our previous responses have addressed your concerns. We are eager to provide further clarification and hope that our subsequent response will fully resolve any remaining issues.
> >
> > **Q3:** The details of the attention mask when ablating Pair Interaction.
> >
> > **Answer:** Thanks for your insightful question. Let us describe the attention mechanism in RelatiViT in detail. We concatenate [image_tokens, subject_query_tokens, object_query_tokens] as a long sequence and input it into the ViT encoder. During the forward pass, the attention between subject_query_tokens and object_query_tokens serves as Pair Interaction.
> >
> > In our ablation study, we mask out the attention weights between subject and object query tokens, which we call “w/o Pair Interaction”. If we denote $n_{I}$ as the number of image tokens, $n_{s}$ as the number of subject query tokens, and $n_{o}$ as the number of object query tokens, our attention map is dimensioned as $attn.shape = (n_{I}+n_{s}+n_{o}, n_{I}+n_{s}+n_{o})$. To ablate pair interaction, we mask out attentions from subject queries to object queries, i.e., $attn[n_{I}:n_{I}+n_{s}, n_{I}+n_{s}:n_{I}+n_{s}+n_{o}]=0$, and vice versa, i.e., $attn[n_{I}+n_{s}:n_{I}+n_{s}+n_{o}, n_{I}:n_{I}+n_{s}]=0$. This approach is somewhat akin to the block matrix you referenced. However, a crucial distinction remains: both subject and object queries continue to attend to the contextual image tokens. This distinguishes our approach from a mere self-attention mechanism with shared weights. We appreciate your suggestion and will enhance the clarity of this description in our revised manuscript.
> >
> >
> > **Q4:** Applying CoordConv to CNNTransformer.
> >
> > **Answer:** Sorry for the misunderstanding. We follow your suggestion to apply CoordConv to CNNTransformer, and the results are presented in the table below. It is evident that even with the incorporation of spatial positional embeddings, CNNTransformer's performance remains worse than RelatiViT. This result suggests that the positional embeddings offer minimal benefit to CNNTransformer. Given this controlled experiment, we can infer that CNN backbones are less effective in extracting spatial relation information compared with transformers. This observation aligns with your earlier comments and further validates our approach.
> >
> > |  Method   |  %Avg. Acc.  | %F1  |
> > | :----: | :----: | :----: |
> > | CNNTransformer-CoordConv  | $68.69\pm1.23$ | $66.47\pm1.81$ |
> > | RelatiViT (ours) | $80.02\pm0.73$ | $78.32\pm1.01$ |
> >
> >
> > **Q5:** Conducting the intervention experiment on CNNTransformer.
> >
> > **Answer:** Thanks for the further question. We would like to gently remind that the primary objective of this experiment is to address your earlier concern regarding our statement that “[prior methods] are prone to simply predict the relationships based on the bounding box coordinates rather than learn the intended solutions by referring to the RGB input” in Section 5.4. Consequently, our experiment compares the sensitivity to image replacement in prior methods (RUNet and DRNet) with our RelatiViT model. The results clearly demonstrate that RelatiViT relies more on visual cues for prediction.
> >
> > Furthermore, in line with your suggestion, we conduct the same intervention experiment with CNNTransformer. As illustrated in the table below, it is evident that the prediction flipping ratios for both CNNTransformer and RelatiViT are comparable to each other, and are notably higher than those for RUNet and DRNet. This result suggests two key insights:
> >
> > 1) Our transformer-centric design principles (including feature extraction, query localization, context aggregation, and pair interaction), which do not incorporate bounding box coordinates into the input unlike previous methods, are indeed more inclined to utilize visual information for predicting spatial relations. This finding supports the rationale behind our design choices, especially in the context of spatial relation prediction tasks.
> >
> > 2) Despite a similar emphasis on visual information, RelatiViT significantly outperforms CNNTransformer. This indicates the superior capability of the vision transformer backbone in extracting spatial relation information compared to CNNs.
> >
> > We hope this clarification further substantiates the effectiveness of our method.
> >
> > | Method | %Prediction flipping ratio |
> > | :--------: | :-------: |
> > | DRNet | $2.33 \pm 2.11$ |
> > | RUNET | $6.99 \pm 2.26$ |
> > | CNNTransformer | $30.18\pm1.69$ |
> > | RelatiViT | $29.47\pm1.88$ |

---

> > > ### Comment · Reviewer_ubwR · 2023-11-22
> > >
> > > Thank you very much for the clarifications and for engaging this much with the reviewing process. I really appreciate it. Overall, all my concerns have been addressed.
> > >
> > > Q3: this is very interesting. I appreciate the additional nuance in clarifying the different attentional components. I think the paper will be much stronger with the additional clarifications.
> > >
> > > Q4: This is also very interesting. I expected the performance here would be higher giving the bounding box performance being so high. It is kind of amazing that the bounding box can do so well when the CNN-transformer does this poorly. You do not have to do anything else for the reviews, but I think that some additional discussion of those results and potential suggestions for experiments to further explore this phenomena would make for a very insightful discussion section.
> > >
> > > Q5: Thanks for running the additional experiments and my apologies for not making the connection between your answer and the weakness I had raised. I was very focused on the comparisons in Table 3 as I thought it was the salient point, and overlooked the other weakness I pointed out in my first review. I think the new experiments really substantiate your work by indicating how much the CNN-Transformer model attends to the visual information as well as its relatively weaker performance.
> > >
> > >
> > > ---------
> > > I first want to emphasize that you do not have to do this experiment; It is just an experiment that I thought might be interesting to consider as it tries to address something that I am still confused about.
> > >
> > > It remains very puzzling to me how bad the CNN+transformer model is given the strong performance of the bbox baseline. The very nice experiment you report in Q5's answer sheds some light on this: the model is paying attention to visual features which may be less informative, and as a result, might be overfitting to specific visual patterns in the data and failing to generalize.
> > >
> > > One experiment that could shed more light on this is replacing the CNN features with ONLY the positional embeddings for a PosEmbed+Transformer baseline. This baseline would not depend on any visual information and at a high-level, should be learning something equivalent to the bbox baseline as it is getting the same information with a different parameterization.
> > >
> > > It would be very interesting to see its performance. If it performs as well as the bbox baseline, it would further support the claim the CNN's features are bad and potentially resulting in overfitting (this should be accompanied with CNN+transformer also having a better training set performance). On the other hand, it might be possible that this baseline does worse or as well as the CNN+transformer. In this case, it would be very interesting to understand what about the transformer and the positional-encoding parametrization is resulting in this issue. I might be missing some subtlety or detail that makes the PosEmbed+Transformer baseline different from the bbox baseline, but it seems at a high level that they should learn equivalent functions and hence achieve similar performance.

---

> > > > ### Author Response · Authors · 2023-11-23
> > > >
> > > > Thank you very much for your insightful and constructive feedback on our manuscript. We sincerely appreciate the time and effort you have devoted to reviewing our work.
> > > >
> > > > Your suggestion to include a discussion section that delves deeper into the superior capabilities of transformers compared to CNNs is well-received. We agree that this addition will significantly enhance the depth and clarity of our paper. We will incorporate this section in our final version and ensure it thoroughly explores the relevant aspects as per your recommendation.
> > > >
> > > > Once again, we are grateful for your valuable comments.
> > > >
> > > > Wishing you all the best,
> > > >
> > > > Authors

---

### Official Review · Reviewer_hAXs · 2023-10-28

**Soundness:** 3 good
**Presentation:** 3 good
**Contribution:** 2 fair
**Rating:** 5
**Confidence:** 4

**Summary:**

The paper investigates the ability of current computer vision systems to recognize physically grounded spatial relations between objects. The authors propose precise relation definitions that allow for consistent annotation of a benchmark dataset and find that existing approaches perform poorly on this task. They propose new approaches that exploit the long-range attention capabilities of transformers and evaluate key design principles. The authors identify a simple "RelatiViT" architecture that outperforms all current approaches and demonstrate that it is the first method to convincingly outperform naive baselines on spatial relation prediction in in-the-wild settings. The findings have important implications for the development of computer vision systems and their ability to recognize spatial relations.

**Strengths:**

1. The paper proposes precise relation definitions that permit consistently annotating a benchmark dataset.
2. the paper identifies a simple "RelatiViT" architecture that outperforms all current approaches and convincingly outperforms naive baselines on spatial relation prediction in in-the-wild settings.
3. The paper evaluates key design principles and provides insights into the effectiveness of different approaches for recognizing physically grounded spatial relations between objects.

**Weaknesses:**

1. The spatial relationships examined in this paper are rather straightforward, as they do not encompass interactions between humans and objects or comparative relationships among multiple objects. This limitation might confine the applicability of the proposed method.
2. This paper would benefit from a more extensive examination of the reasons behind the superior performance of "RelatiViT" compared to the "bbox-only" baseline. Current methods utilizing image features do not match the simplicity and effectiveness of the "bbox-only" baseline. What sets "RelatiViT" apart?
3. As illustrated in Table 6, "RelatiViT" does not consistently outperform the "bbox-only" baseline across various categories of spatial relations. The potential advantages of "RelatiViT" might not be evident when taking into account the higher computational demands it imposes for image feature processing.
4. The paper fails to elucidate the reasons why "RelatiViT" exhibits superior performance in the "left" category as opposed to the "bbox-only" baseline, while displaying subpar performance in the "right" category.
5. Although it is good to unveil that the current methods and datasets have limitations on this new task, it is still imperative to do a thorough study on each proposed component to show why such a design is essential to the performance improvements. It is hard to be convinced about the significance of the contribution based on the current form.

**Questions:**

Please refer to the weakness part.

---

> ### Author Response · Authors · 2023-11-21
>
> Thanks for your valuable feedback. We hope the following response will address your concerns.
>
> **Q1:** The applicability of the proposed method is limited because the spatial relations do not include the human-object and multi-object relations.
>
> **Answer:** Human-object interaction and multi-object relations are indeed promising topics to investigate. However, we find the existing methods struggle to accurately resolve even the fundamental spatial relations addressed in our paper. In this paper, we aim to investigate this spatial relation prediction task to show the importance and challenges of this under-explored field. Our intention is to provide a foundational framework that can serve as a springboard for tackling more complex scenarios in future research.
>
> As for the applicability, our method can be easily applied to the robot systems to query for spatial relations. This capability can be further leveraged for advanced applications such as PDDL-based planning or unsupervised reward learning, which will boost the performance of robot learning.
>
> **Q2:** This paper would benefit from more analysis of the superior performance of RelatiViT and a thorough study on each proposed component.
>
> **Answer:** Thanks for your kind suggestion and for recognizing the good performance of our method. As also mentioned by Reviewer XTRZ and Reviewer ubwR, we have provided extensive experiments to analyze the success of RelatiViT in Section 5.5, Appendix A5, A7 and A8. In Section 5.5, we provide an in-depth ablation study, revealing that both context aggregation and good feature extraction are key contributors to the performance. The attention maps in Section 5.5 and Appendix A8 demonstrate that RelatiViT attends to appropriate regions to aggregate the context information for accurate spatial relation prediction. And the comparisons on each relation category in Section 5.5 and Appendix A5 show that RelaTiViT significantly surpasses baselines (especially bbox-only) in categories where visual information is critical. This indicates that RelatiViT successfully extracts the visual representations containing rich spatial relation information. Additionally, the prediction visualizations in Appendix A7 showcase that RelatiViT predicts the relations based on the visual information, e.g., object contact, occlusion, depth and background context, etc. This capability fundamentally contributes to its superiority over bbox-only baselines.
>
> **Q3:** "RelatiViT" does not consistently outperform the "bbox-only" baseline across various categories of spatial relations.
>
> **Answer:** In Table 6, our results are either comparable or outperform the bbox-only baselines across most categories. Moreover, in Table 10, RelatiViT performs better than bbox-only on 29 out of 30 predicate categories, which is a significant improvement.
>
> It is essential to recognize that while bbox-only models may achieve comparable accuracy in certain scenarios, such blind models, which do not leverage image inputs, possess inherent limits in further improvements. In contrast, our method leverages both visual data and advanced attention mechanisms, marking a notable advancement in predicting spatial relations. For the first time, our work demonstrates that vision-based methods can substantially exceed the bbox-only baseline. This achievement highlights the pivotal role of visual cues and relation modeling in this task. We believe that future research building upon our methodology will achieve even greater accuracy, far surpassing simpler baselines.
>
> **Q4:** The reasons why "RelatiViT" exhibits superior performance in "left" but worse in “right”.
>
> **Answer:** There might be some misunderstanding. According to Table 6, the performance of RelatiViT is comparable with bbox-only on “left” and “right”. However, it's important to highlight that RelatiViT demonstrates significantly superior performance on more complex spatial relations such as "behind", "in", "under", and "front". These particular relations inherently require a more nuanced understanding of visual information.
>
> Thanks again for your precious time. Please let us know if there are any additional questions.

---

> > ### Comment · Reviewer_hAXs · 2023-11-22
> >
> > Thanks for addressing my raised issues. I decided to change my rating to marginally above the acceptance threshold.

---

> > > ### Author Response · Authors · 2023-11-22
> > >
> > > We are pleased to see that your concerns have been addressed and deeply appreciate your willingness to consider improving your score.
> > >
> > > We wanted to ensure that you have the opportunity to update your rating. As of now, we still observe your rating as a 5. Please feel free to reach out if there are any new concerns or further feedback you would like to share with us.

---

### Official Review · Reviewer_MUvq · 2023-10-30

**Soundness:** 3 good
**Presentation:** 3 good
**Contribution:** 2 fair
**Rating:** 6
**Confidence:** 4

**Summary:**

This paper sets out to study transformers' capability of understanding spatial relations in a scene. The authors strictly define spatial relations and refine previous datasets accordingly. The authors propose several transformer-based models to tackle spatial relation prediction task.

**Strengths:**

- It is interesting to delve into whether transformers could understand spatial relations in a scene instead of simply exploiting spatial biases.

- The linguistic biases during annotation brought up by this paper are reasonable.

- The proposed models are reported to be effective on refined datasets. The discussion on design axes is informative.

- Attention map visualization is insightful as it shows that the proposed model captures meaningful contextual information when making a prediction.

**Weaknesses:**

- Although resolving linguistic biases in annotations is a well-motivated aspect of this work, I am not sure if these issues are nontrivial. It would be great if there were a quantitative bias analysis of previous datasets.
- Besides linguistic biases during the annotation process, the statistics of spatial relations are biased, which could lead spatial relation detection to simply object classification. Rel3D addressed it by employing minimally contrastive construction, I would like to see authors' discussion on SpatialSense+.
- Despite being titled as "Can Transformers Capture Spatial Relations between Objects", all prior baselines are CNN-based methods, it would be great to see authors include more advanced transformer-based models [1].

[1] SGTR: End-to-end Scene Graph Generation with Transformer, Li et al, 2021.

**Questions:**

Please see weaknesses above.

---

> ### Author Response · Authors · 2023-11-21
>
> Thanks for your valuable suggestions. We hope the following response will address your concerns.
>
> **Q1:** Quantitative bias analysis of linguistic biases.
>
> **Answer:** To demonstrate quantitatively how linguistic bias affects annotation quality, we analyze the top 20 subject-predicate-object triplets with the most significant change in labels after our relabeling process in the SpatialSense dataset. We also provide the issue types that make the original annotation incorrect, including object-centric, polysemous words, and idiomatic expressions, i.e., the three types of errors as discussed in Section 3.2. Our findings reveal that idiomatic expressions account for 20% of the incorrectly labeled triplets, polysemous words contribute to 10% of these errors, and the remaining 70% is mixing up object-centric and viewer-centric relations. The percentages of idiomatic expressions and polysemous words highlight the typical failure cases caused by linguistic bias in the dataset.
>
> | Triplet | Label Flip Ratio | Type |
> | :----: | :----: | :----: |
> | window-in front of-chair | 1.0 | object-centric |
> | frame-on-wall | 1.0 | idiomatic expressions |
> | women-to the left of-women | 1.0 | object-centric |
> | girl-to the left of-man | 1.0 | object-centric |
> | bus-in-road | 1.0 | idiomatic expressions |
> | book-on-shelf | 0.95 | polysemous words |
> | books-on-cupboard | 0.8 | polysemous words |
> | woman-to the left of-man | 0.8 | object-centric |
> | man-to the left of-woman | 0.8 | object-centric |
> | woman-to the left of-woman | 0.78 | object-centric |
> | table-in front of-chair | 0.75 | object-centric |
> | boy-to the left of-girl | 0.75 | object-centric |
> | woman-to the right of-woman | 0.75 | object-centric |
> | chair-to the right of-chair | 0.69 | object-centric |
> | man-on-wall | 0.67 | idiomatic expressions |
> | tree-in-ground | 0.67 | idiomatic expressions |
> | chair-next to-desk | 0.67 | object-centric |
> | women-to the left of-men | 0.67 | object-centric |
> | building-next to-tree | 0.67 | incorrect definition |
> | hen-to the left of-hen | 0.67 | object-centric |
>
>
> **Q2:** Discussion about the statistics bias of spatial relations.
>
> **Answer:** Thanks for this insightful question. The original SpatialSense dataset is collected with adversarial crowdsourcing strategy. Specifically, the authors trained a simple model to predict spatial relations mainly based on the statistics, and then they asked crowdsourcing annotators to only label the samples that cannot be handled by this model. In this process, the statistics bias has been much alleviated. Our SpatialSense+ retains these samples and predicate categories from the original dataset. Consequently, we inherit this advantage of mitigating statistical bias.
>
> **Q3:** Compare with more advanced transformer-based models.
>
> **Answer:** Thank you for your insightful suggestions. We apologize for missing references such as SGTR [1] and RelTR [2]. Firstly, we want to clarify that these works are in different settings from ours: SGTR and RelTR aim to generate scene graphs and include an end-to-end detection process, but in our setting the detection is not required. Recognizing the distinct settings of our work, we make much effort to adapt RelTR to align with our experimental setting. This involved maintaining the same ResNet backbone as RelTR, encoding bounding box and object category information into learnable entity tokens, and modifying the relation prediction head to mirror the K-way binary classification head used in our models.
>
> Upon implementing these changes, we observed an average accuracy of 57.91% with RelTR, which is markedly lower than anticipated. We attribute this underperformance to two primary factors:
>
> (1) RelTR relies on the features from only the last layer of ResNet, which we believe results in a significant loss of spatial information.
>
> (2) RelTR leverages learnable tokens to establish correspondence between the queries and the feature in region of interests, and then aggregate the information. This is difficult to achieve by the supervision of a small training set. Differently, our model utilizing large-scale pretrained transformer features and design a mask image based query localization strategy, which is able to establish the correspondence with fewshot training samples.
>
> This experiment further highlights the capability to capture spatial relationships of our model. We will definitely include these references in our revision.
>
> [1] Li R, Zhang S, He X. Sgtr: End-to-end scene graph generation with transformer[C]//proceedings of the IEEE/CVF conference on computer vision and pattern recognition. 2022: 19486-19496.
>
> [2] Cong Y, Yang M Y, Rosenhahn B. Reltr: Relation transformer for scene graph generation[J]. IEEE Transactions on Pattern Analysis and Machine Intelligence, 2023.
>
> We appreciate your precious time in reviewing our paper. Please let us know if you have additional questions.

---

> > ### Comment · Area_Chair_oaA8 · 2023-11-23
> >
> > Dear Reviewer,
> >
> > The author has provided responses to your questions and concerns. Could you please read their responses and ask any follow-up questions, if any?
> >
> > Thank you!

---

> ### Author Response · Authors · 2023-11-23
> **Has our response addressed your concerns?**
>
> Dear Reviewer,
>
> Thanks again for your precious time and insightful suggestions. We hope we have addressed your concerns comprehensively. As the deadline is approaching, with less than 8 hours remaining, we kindly request that you let us know if there are any further questions or clarifications needed.
>
> Best regards,
>
> Authors

---

> ### Comment · Reviewer_MUvq · 2023-11-23
>
> Thanks for the authors' responses. I think my concerns have been addressed. I also found the discussions of **hAXs.1**, **ubwR.3**,  **ubwR.5**,  **Jenr.2**,  **Jenr.5** valuable. Thus, I would like to raise my score to 6 and would not object to accepting this paper.

---

> > ### Author Response · Authors · 2023-11-23
> >
> > Thank you very much for your valuable feedback and willingness to consider improving your score!
> >
> > Wishing you all the best,
> >
> > Authors

---

### Official Review · Reviewer_XTRZ · 2023-10-31

**Soundness:** 4 excellent
**Presentation:** 4 excellent
**Contribution:** 3 good
**Rating:** 8
**Confidence:** 4

**Summary:**

The paper studies how to understand spatial relationship between objects within a single image. GIven that the current benchmarks are synthetic (Rel3D) or ambiguous (SpatialSense), it starts with a re-annotation of SpatialSense with precise relation definitions and calls the new dataset SpatialSense+. Since existing state-of-the-art methods perform poorly on this benchmark, the paper proposes a RelatiViT, which is a ViT based network to resolve the problem. This is the first approach which outperforms naive baseline significantly on multiple benchmarks.

**Strengths:**

Overall, I think this is a solid paper, and recommend accepting the paper.

- The paper discusses limitations of current spatial relationship benchmarks. That makes a lot of sense to me.
- The paper did an extensive study of different network architectures to capture the spatial relationship between objects.
- Experiments suggest the proposed RelatiViT has strong performance on multiple benchmarks.
- The writing is clear and easy to follow.

**Weaknesses:**

- The paper mainly compared RelatiViT with ConvNet and ViT based networks. However, some of current multimodal Large language models (e.g. LLaVA [1], or even GPT-4V) seem be to capable of handling some spatial relationships. But I think it's acceptable to ignore these literature as they are very recent works and are not specially designed for this task.
- The annotation of SpatialSense+ can be very hard to scale, since it requires annotators to understand precise definition of spatial relationships.


[1] Haotian Liu, Chunyuan Li, Qingyang Wu, Yong Jae Lee. Visual Instruction Tuning. NeurIPS 2023.

**Questions:**

See weaknesses.

---

> ### Author Response · Authors · 2023-11-21
>
> Thank you very much for the insightful suggestions and supportive comments. We hope the following response will address your concerns.
>
> **Q1:** Compare with large vision language models.
>
> **Answer:** Thank you for noting that our work precedes the publicly available large vision-language foundation models. It is indeed valuable to evaluate these models on spatial relationship prediction extensively, and we have performed some early tests on LLaVA. We evaluate the SpatialSense+ dataset on LLaVA latest version, i.e., LLaVA-v1.5-13b, and it only gets 50.27% overall accuracy, indicating that LLaVA   cannot handle this physically grounded spatial relation task. We think this result is very interesting and provides some inspiration for us about future work, so thank you very much again for this insightful suggestion.
>
> Regardless of the specific results, for the foreseeable future, computation and time costs make it impractical for these foundation models to be directly applicable in many scenarios of interest, such as robotics with size, weight, power, and real-time operation constraints. We believe that it is valuable both practically for these applications and scientifically to understand and maximize the computer vision capabilities of smaller models for a large variety of tasks including ours.
>
> **Q2:** The annotation of SpatialSense+ can be very hard to scale.
>
> **Answer:** Thanks for your insightful question. We do not expect the labeling process to scale up, but want to fix this issue by taking advantage of large-scale unsupervised data and few-shot labeled data (i.e., self-supervised pretraining and finetuning). We admit that annotating such complex tasks cannot easily scale up, because it would definitely be very difficult and cost a lot of labor and financial resources. However, with the recent progress of foundation models, finetuning from large pretrained models makes it possible to inherit rich prior knowledge to handle complex downstream tasks with few-shot training sets. This has been a common solution for difficult problems in deep learning nowadays. Motivated by this, in this paper, we aim to fine-tune a good relation prediction model from the pretrained models with a few samples. In the future, especially in the age of large foundation models, we expect the solution to be reading out the relation information from foundation models by few-shot finetuning or even zero-shot prompting. Therefore, we do not require a large-scale spatial relation dataset, but only need a good pretrained model and a well-designed finetuning architecture and strategy, which is actually what we investigate in this paper.
>
> Once again, we are grateful for the valuable and supportive comments!

---

> > ### Comment · Reviewer_XTRZ · 2023-11-22
> > **Re: Official Comment by Authors**
> >
> > I appreciate your LLaVA experiments. I personally feel it sometimes gets the correct spatial relationship, but I have never tested it on any benchmark.
> >
> > I'm confident to keep my rating after reading other reviews and your response.

---

> > > ### Author Response · Authors · 2023-11-22
> > >
> > > Thank you very much for your precious time and valuable feedback!

---

### Official Review · Reviewer_Jenr · 2023-11-01

**Soundness:** 3 good
**Presentation:** 4 excellent
**Contribution:** 2 fair
**Rating:** 5
**Confidence:** 4

**Summary:**

This paper investigates the task of spatial relation classification on a single 2D image. Given two objects bounding boxes, the authors uses the nine predicates proposed in the SpatialSense dataset and correct some of its ambiguous labels. To solve the task, the authors present different architectures and promote a simple transformer based model. Experiments are also conducted on REL3D dataset with ablation studies.

**Strengths:**

The authors propose a more accurate definition of the problem of spatial relation, although the limitations of this definition is not completely covered.

The motivation to use transformers to model object relation is sensible, as relation modelling this is one of the main motivations over CNNs baselines.

The proposed architecture is validated with different baselines and ablation studies. In particular, I appreciate that the author also report the results of naive baselines. Experiments confirmed the usefulness of the proposed transformer mechanism to model relations.

**Weaknesses:**

The current superiority of backbone transformers over CNNs and the ability of transformers to model object relations are commonly admitted. So in this aspect, this downgrade the insights brought by the results of the paper.

The claim of the authors to provide "a precise and unambiguous definition" seems a bit exaggerated. There are natural ambiguities in the task of spatial relationships. For instance, even for humans, it is difficult to judge on same natural images the relations "above" if the objects are very distant and that it is not easy to judge if they are on the same plane. For instance, between a car and a mountain in the background (this is one of the examples provided in the supplementary material).

The definition provided by the authors include subjective (which seems naturally inevitable for this task) appreciations in the supplementary material. For instance, the relation "behind" is ignored if the objects are "in different directions" and the relation "in front of" is ignored "if the distance is too large". This definition is not provided by the author, and in practice, it might even depend on the application and the image context. In the case of robotic application, which is the motivation suggested in the introduction of the paper, the definition of "next" could applied to all objects belonging to the same group.

Lastly, the author reuse the same predicates as the previous dataset. This limits their contribution to making corrections on an existing dataset.

In general, the paper does not bring a lot of new insights about the method as the modelling power of transformer is well known. This work do bring an improvement to an existing dataset, enabling researchers to better evaluate their method in the task of spatial relations, but I am not sure whether this in itself would be of interest to the ICLR community. Maybe this paper would be a better fit for a more vision or robotic dedicated conference.

**Questions:**

in section 5.4, the author wrote "bbox-language cannot beat bbox-only, indicating that we have successfully removed the language bias in our benchmark."

However, in Table 11 where there is an evaluation on the original SpatialSense, where the bbox-only is also superior to bbox-language. Then, how the author can conclude to the relations between these two baselines and the language bias ?

---

> ### Author Response · Authors · 2023-11-21
>
> Thanks for your valuable suggestions. We hope the following response will address your concerns.
>
> **Q1:** The proposed transformer-based method does not bring a lot of new insights because the superiority of transformer is commonly admitted.
>
> **Answer:** Firstly, we want to restate that our argument in this paper is not only transformer is better than CNN, but also transformer architectures are not always optimal but need special design and correct visual prompts to read out the relation information, as we compared and discussed in  Section 5.3. Besides, transformers actually have limited improvements on common tasks, e.g., image classification (top1 acc: ImageNet ResNet101 80.67 V.S. ViT-Base 80.73) and detection (COCO AP: Faster RCNN 42.0 V.S. DETR 43.3). In contrast, our transformer architecture shows significant performance boosting on the spatial relation task, as also mentioned by Reviewer XTRZ, MUvq and ubwR. We believe the architectural design principles will provide insight for the community.
>
> **Q2:** There are natural ambiguities in the task of spatial relationships. So the claim of "a precise and unambiguous definition" seems a bit exaggerated.
>
> **Answer:** A precise definition of the spatial relationship is possible with complete knowledge of the scene. This is different from whether a 2D image of the scene always presents this information unambiguously. This is also true for other types of visual properties that we like to infer, such as object shape: humans may not always be able to infer full 3D shapes of objects from an image, but the 3D shape is of course well-defined. Therefore, we provide precise and unambiguous definitions according to the geometries to annotate the dataset. As for the capability of vision model to solve this challenging task based on 2D images where there might existing ambiguity, we think it can be solved by large-scale pretraining and special architecture design. In this paper, we demonstrate this potential solution by our ReltiViT and IBOT pretrained vision transformer. We think this is a good starting point.
>
> **Q3:** The definitions provided by the authors include subjective (which seems naturally inevitable for this task) appreciation.
>
> **Answer:** Thank you for raising this point. It is indeed more precise to separate an "in-principle definition" from an "operationalizable definition" that an annotator can use to provide consistent annotations. We will clarify that these definitions are intended to focus human labeling efforts on unambiguously true predicates. In our annotation process, a sample was annotated by several annotators and the final label was voted among these multiple annotations, which reduces this problem.
>
> (This response is continued in the next comment block.)

---

> ### Author Response · Authors · 2023-11-21
>
> **Q4:** The contribution to dataset is limited because the author reuses the same predicates as the previous dataset.
>
> **Answer:** We would like to clarify that the main problem with the original SpatialSense dataset for studying spatial relationship prediction is not that it lacks some predicates, or that it has an uninteresting image distribution, but that its predicate labels are not consistently annotated. This is naturally what we have aimed to fix in our paper. Our experiments show that the relabeled SpatialSense+ is a reliable benchmark to evaluate the spatial relation prediction models, indicating our contribution to the whole community.
>
> **Q5:** The conclusion that language bias is removed cannot be drawn from the experiment result that Bbox-only is better than Bbox+language.
>
> **Answer:** We are sorry for the incorrect statement. There are two kinds of language bias:
>
> (1) The language bias from commonsense: a computer is usually *on* a table, so the model will predict “on” without adequately considering the image and bounding box information, even when the computer is actually under the table. Actually, this kind of language bias has been mitigated by the original SpatialSense dataset through *adversarial crowdsource* strategy. And our SpatialSense+ is relabeled based on the original dataset and thus inherits this property. Consequently, bbox+language performs worse than bbox-only in both SpatialSense+ and SpatialSense datasets. We will correct this point and ensure clarity in our revision.
>
> (2) The language bias from some idiomatic expressions, i.e., the problem we discussed in Section 3.2, which will lead to ambiguous and noisy annotation, e.g., the boy is sometimes labeled as *in* snow and sometimes *on snow*. To illustrate this, we show some examples in the tables below, where language biases impact labeling. For example, “book-on-shelf” samples are actually False because the books were not placed *on* the top of the shelves, but *in* one of the tiers of the shelf. These samples cannot be predicted only according to language and only affect the annotation quality. During our relabeling process, these mistakes are corrected by our precision and physically grounded definitions. These examples underscore our success in eliminating idiomatic expression language bias. We will include this analysis in our revision.
>
> **On:**
> | Triplet | Label Flip Ratio |
> | :----: | :----: |
> | frame-on-wall | 1.0 |
> | book-on-shelf | 0.95 |
> | books-on-cupboard | 0.8 |
> | man-on-wall | 0.67 |
> | sign-on-building | 0.33 |
>
> **In:**
> | Triplet | Label Flip Ratio |
> | :----: | :----: |
> | bus-in-road | 1.0 |
> | tree-in-ground | 0.67 |
> | dog-in-water | 0.33 |
> | man-in-water | 0.33 |
> | wire-in-wall | 0.25 |
>
> We appreciate your precious time. We are eager to engage in further discussions to clear out any confusion.

---

> ### Author Response · Authors · 2023-11-23
> **Has our response addressed your concerns?**
>
> Dear Reviewer,
>
> Thanks again for your precious time and insightful suggestions. We hope we have addressed your concerns comprehensively. As the deadline is approaching, with less than 8 hours remaining, we kindly request that you let us know if there are any further questions or clarifications needed.
>
> Best regards,
>
> Authors

---

### Meta-Review · Area_Chair_oaA8 · 2023-12-06

**Metareview:**

The paper proposes a systematic framework for studying spatial relationships. This includes precise relationship definitions, a benchmark dataset, and a new approach.

After reading the reviews and the paper, AC recommends acceptance with a poster. AC recommends the authors address all the feedback from the reviewers in the final version. This includes clarifications on spatial relationship definitions and ambiguities in different contexts raised by reviewer Jenr.

Strengths and weaknesses are summarized below:

Strengths:
1. the paper is well-written
2. the definitions, motivations, and limitations of existing works are highlighted clearly
3. the performance of the proposed approach is much better than existing approaches
4. the paper contributes a systematic framework and a new benchmark dataset to study spatial relationships.

Weakness:
1. the paper can be improved by clarifying the definitions of spatial relationships (reviewer Jenr)
2. the paper can report the performance of large language models in such tasks (reviewer XTRZ)
3. A discussion on dataset biases and human annotation biases is needed (reviewers MUvq, ubwR)

**Justification For Why Not Higher Score:**

The paper proposes an interesting dataset studying spatial relationships. The work is interesting to the community.
However, the contributions are not sufficient to be accepted as spotlight/oral.

**Justification For Why Not Lower Score:**

The paper studies a fundamental problem of spatial relationship understanding. The insights obtained are interesting. The paper also contributes a useful dataset. Overall, the strengths outweigh the weaknesses. Hence, AC recommends (accepting with a poster).

---

### Decision · Program_Chairs · 2024-01-16

Accept (poster)